# Fine-tuning spatial-temporal dynamics and surface receptor expression support plasma cell-intrinsic longevity

Zhixin Jing[1], Phillip Galbo[2,3], Luis Ovando[1], Megan Demouth[2], Skylar Welte[1], Rosa Park[1], Kartik Chandran[3], Yinghao Wu[3,4], Thomas MacCarthy[5], Deyou Zheng[2,4], David Fooksman[1,2]*

[1]Department of Pathology, Albert Einstein College of Medicine, Bronx, United States; [2]Department of Genetics, Albert Einstein College of Medicine, Bronx, United States; [3]Department of Microbiology and Immunology, Albert Einstein College of Medicine, Bronx, United States; [4]Department of System and Computational Biology, Albert Einstein College of Medicine, Bronx, United States; [5]Department of Applied Mathematics and Statistics, Stony Brook University, Stony Brook, United States

*For correspondence:
david.fooksman@einsteinmed.
edu

Competing interest: The authors declare that no competing interests exist.

**Abstract** Durable serological memory following vaccination is critically dependent on the production and survival of long-lived plasma cells (LLPCs). Yet, the factors that control LLPC specification and survival remain poorly resolved. Using intravital two-photon imaging, we find that in contrast to most plasma cells (PCs) in the bone marrow (BM), LLPCs are uniquely sessile and organized into clusters that are dependent on APRIL, an important survival factor. Using deep, bulk RNA sequencing, and surface protein flow-based phenotyping, we find that LLPCs express a unique transcriptome and phenotype compared to bulk PCs, fine-tuning expression of key cell surface molecules, CD93, CD81, CXCR4, CD326, CD44, and CD48, important for adhesion and homing. Conditional deletion of *Cxcr4* in PCs following immunization leads to rapid mobilization from the BM, reduced survival of antigen-specific PCs, and ultimately accelerated decay of antibody titer. In naïve mice, the endogenous LLPCs BCR repertoire exhibits reduced diversity, reduced somatic mutations, and increased public clones and IgM isotypes, particularly in young mice, suggesting LLPC specification is non-random. As mice age, the BM PC compartment becomes enriched in LLPCs, which may outcompete and limit entry of new PCs into the LLPC niche and pool.

## eLife assessment

Despite the importance of long-lived plasma cells (LLPCs), particularly for the infection and vaccination field, it is still unclear how they acquire their longevity. With a **solid** genetic approach, the authors demonstrate quite **convincingly** a requirement for chemokine/chemokine receptor-mediated interaction in LLPC longevity. The data are very **valuable** for the development of new types of vaccines.

## Introduction

Prophylactic antibodies induced by vaccines provide rapid, systemic and in some cases, long-lasting immune protection against many infectious diseases. Variability in the duration of antibody responses is chiefly dependent on the composition of short-lived and long-lived plasma cells (LLPCs) produced, which can have distinct lifespans of a few days and months or years in mice, respectively (*Sze et al., 2000*). The ability to generate LLPCs also declines with old age, and hence the durability of the

vaccine response (*Frasca and Blomberg, 2020*; *Palacios-Pedrero et al., 2021*). Therefore, understanding how LLPCs are generated and maintained are essential for enhancing durability of vaccine-induced antibody responses in humans.

Previous studies have reported these LLPCs are enriched in the bone marrow (BM) but can also be found in spleen and mucosa (*Manz et al., 1997*; *Slifka et al., 1998*; *Bortnick et al., 2012*; *Bohannon et al., 2016*; *Lemke et al., 2016*). Tracking of endogenous polyclonal LLPCs is challenging, requiring labeling and tracking by thymidine analogs like BrdU, or looking for antigen-specific antibody-forming cells by ELISPOTs. However, these approaches are not amenable to tracking live cells by flow cytometry, as there have been no phenotypic markers for endogenous LLPCs, making these cells elusive. Approaches to genetically track LLPCs were recently established (*Xu et al., 2020*), which allows studying their turnover and generation.

One major question is how these cells are specified. LLPCs can mature from newly minted plasmablasts in the germinal center that have undergone affinity maturation (*Takahashi et al., 1998*; *Phan et al., 2006*). However, LLPCs can also develop in a T cell-independent manner (*Bortnick and Allman, 2013*) and B-1 lineages (*Vergani et al., 2022*), suggesting that there are multiple, distinct pathways to becoming LLPCs, or specification is regulated extrinsically by their niche, or both (*Robinson et al., 2020*). Thus, it is unclear if LLPCs arise from unique clones, unique pools of B cells, or are just randomly specified from the bulk plasma cell (PC) pool, in a stochastic manner, potentially through maturation in the bone marrow (BM) niche. Determining what is required for LLPC specification is important for vaccine development.

A second major question regarding LLPCs is how they are maintained and survive in a cell-specific manner. While functionally they are metabolically active, quiescent, murine LLPCs (defined as B220-2NBDG$^+$) are thought to have minimal transcriptional specificity compared to bulk PCs (*Lam et al., 2018*). In contrast, human LLPCs (CD19$^-$ CD138$^{high}$) have been shown to be transcriptionally distinct from other mature PCs (*Joyner et al., 2022*).

The BM is a major lodging site for LLPCs and it is believed that key cell-extrinsic cellular and molecular factors support their longevity. PCs' migratory behavior and positioning within BM parenchyma is also linked to chemokine receptor signaling, cell adhesion, cytokine, and age of mice. Previous work from our laboratory found that as PCs age, CXCR4 expression is increased, suggesting LLPCs may upregulate certain key molecules for survival (*Benet et al., 2021*). CXCR4 is a master chemokine receptor for BM tropism but its role in humoral immunity is thought to be dispensable (*Nie et al., 2004*). However, CXCR4 drives PC motility in the BM and is upregulated on PC with aging (*Benet et al., 2021*). We and others have shown that BM PCs are spatially organized in clusters (*Benet et al., 2021*; *Mokhtari et al., 2015*) and PCs are less motile when they enter these clusters, suggesting extrinsic signals may be important cues for motility. Moreover, in mice lacking APRIL, a key survival cytokine for PCs (*Benson et al., 2008*), these clusters were reduced, suggesting clusters and cell dynamics may be functionally important for PC survival.

In this study, we aim to understand what unique features are associated with LLPC physiology, at a molecular, cellular, and spatial-temporal level using cell fate labeling of PCs. We find that these cells exhibit intrinsic changes in gene expression and cell motility patterns that may underlie their unique ability to persist for long periods of time, despite potential competition from a continuously evolving PC pool. Among the factors promoting their survival, CXCR4 plays a dominant cell-intrinsic role in promoting LLPCs retention and survival and thus, maintaining durability of humoral responses.

## Results

### PC turnover rate decreases with mouse age

We previously reported that in middle-aged mice, PC motility and clustering within the BM and their recirculation capacity was increased, in comparison to young mice (*Benet et al., 2021*). We speculated that these changes in PC dynamics could alter homeostatic PC turnover rates and may also reflect changes in frequency of LLPCs within PC pool with aging. To study LLPC survival mechanisms, we constructed a novel mouse line, Blimp1-ERT2-Cre-TdTomato (BEC), which contains a tamoxifen-inducible cre recombinase (ERT2-cre) and fluorescent reporter TdTomato under the control of the *Prdm1* (BLIMP1) locus (*Figure 1—figure supplement 1A*). We verified that >99% of CD138$^{high}$B220-BM PCs were TdTomato$^+$ (*Figure 1—figure supplement 1B*) and that 94% of TdTomato$^+$ were ASCs

(CD138^High) (*Figure 1A*, *Figure 1—figure supplement 1C*). Tomato expression was about 1.5–2 log higher than Blimp-1 negative cells, similar to expression by other reporters in the *Prdm1* endogenous locus (*Liu et al., 2022*; *Robinson et al., 2022*; *Kallies et al., 2004*). To label and track lifespans of polyclonal PCs under steady-state conditions, we crossed allele BEC with *Rosa26^{LSL-EYFP}* conditional reporter (*Srinivas et al., 2001*) to generate BEC-YFP mouse. Acute treatment with tamoxifen for 3 consecutive days induced robust Cre-mediated recombination and irreversible expression of YFP, comprised of 98% PCs at day 5 (*Figure 1A*) but not in the absence of tamoxifen treatment (*Figure 1B*). Over time, YFP⁺ PCs that survive for months should by definition be bona fide LLPCs. However, in these mice, not all LLPCs would be YFP⁺, as new LLPCs should develop in the unlabeled (YFP⁻) fraction.

To study age-related changes in PC turnover, naïve young (6–8 weeks of age) and middle-aged (20–24 weeks of age) BEC-YFP mice were acutely treated with tamoxifen and tracked over 150 days after treatment (*Figure 1C*). At day 5 post treatment, both age groups had similar frequency of YFP⁺ PCs in the BM (~65%) and in the spleen (~62%) (*Figure 1D*). However, over time, the remaining frequency of YFP⁺ cells of total PCs in young mice were significantly lower than in middle-aged mice in both BM and spleen, indicating more PC turnover in the young mice. At day 150, only 1.8% BM PCs were YFP⁺ in young mice compared to 14% in middle-aged mice, while in spleen, 0.6% YFP⁺ PCs remained in young mice compared to 3.1% in middle-aged mice (*Figure 1E and F*). Based on absolute numbers of YFP⁺ PCs, we analyzed rate of PC decay in the BM and spleen and found that BM PCs decay more rapidly in young mice ($t_{1/2}$ = 58 days) than in middle-aged mice ($t_{1/2}$ = 93 days) (*Figure 1G*). In the spleen, decline of labeled PCs was overall faster than in the BM, in line with previous reports (*Xu et al., 2020*). However, we observed that the decline was slightly more rapid in young mice ($t_{1/2}$ = 28 days) as compared to middle-aged mice ($t_{1/2}$ = 39 days) (*Figure 1H*). Thus, we conclude that homeostatic PC turnover is dependent on tissue-specific microenvironment and aging, suggesting that LLPCs may accumulate with aging, particularly in the BM.

## BM LLPCs display cell-intrinsic arrest and clustering

Reduced PC turnover with age, specifically in the BM niche, suggested that PCs were more sessile and better retained in the BM with aging. However, our previous study that showed middle-aged mice had increased overall PC motility and recirculation compared to young mice (*Benet et al., 2021*). We hypothesized that in our previous study, imaging and recirculation measurements did not discriminate between behaviors of LLPCs and immature PCs, which may have different dynamics.

To test this idea, we applied BEC fate labeling to specifically track polyclonal LLPC dynamics and organization in the BM of unimmunized mice. While YFP expression from *Rosa26* reporter was bright enough to visualize labeled LLPCs, YFP⁻ bulk PCs, which also expressed low levels of Tomato from expression of the BEC allele (Tomato^dim), were insufficiently labeled for deep imaging in the BM. Thus, we bred double PC reporter, Blimp1-YFP BEC *Rosa26^{LSL-Tomato}* mice, in which all PCs were YFP^high from expression of the Blimp1-YFP reporter, and with tamoxifen treatment, could be fate-labeled to co-express high levels of Tomato. We treated these mice with tamoxifen and analyzed surface phenotype of PCs at day 5 and 60 post treatment (*Figure 2A*). Tomato^bright labeled PCs were easily discernable from Tomato^dim bulk PCs by flow cytometry (*Figure 2B*). While at day 5 post treatment, Tomato^bright and Tomato^dim PCs were similar in PC maturation markers CXCR4 and CD93, by day 60, Tomato^bright were phenotypically distinct suggesting they had matured to an LLPC state (explored further in the next section). Using intravital time-lapse imaging, we compared Tomato^bright and Tomato^dim BM PCs dynamics in young and middle-aged mice at day 5, day 30, and day 60 post tamoxifen in order to determine the contribution of intrinsic PC age/maturity to their motility and positioning. We could discriminate and track both PC populations on the basis of Tomato expression in the same time-lapse movies (*Figure 2C*, *Figure 2—video 1*). At day 5 after treatment, dynamics of both subsets of PCs were similar, based on cell trajectories, track and displacement velocities, and mean-squared displacement analysis (*Figure 2D*). However, at day 30 and 60 timepoints, Tomato^bright PCs showed reduced motility as compared to Tomato^dim PCs indicating PC age correlated with reduced cell motility. This effect with PC aging was seen in both young and middle-aged mice, suggesting it was cell intrinsic, and thus related to LLPC maturation. While average speeds for Tomato^bright PCs were relatively slow, some rare PCs were highly motile. At day 30 and 60 timepoints, these fast cells were predominantly in Tomato^dim populations (*Figure 2E*), consistent with immature, short-lived PCs having faster motility than LLPCs.

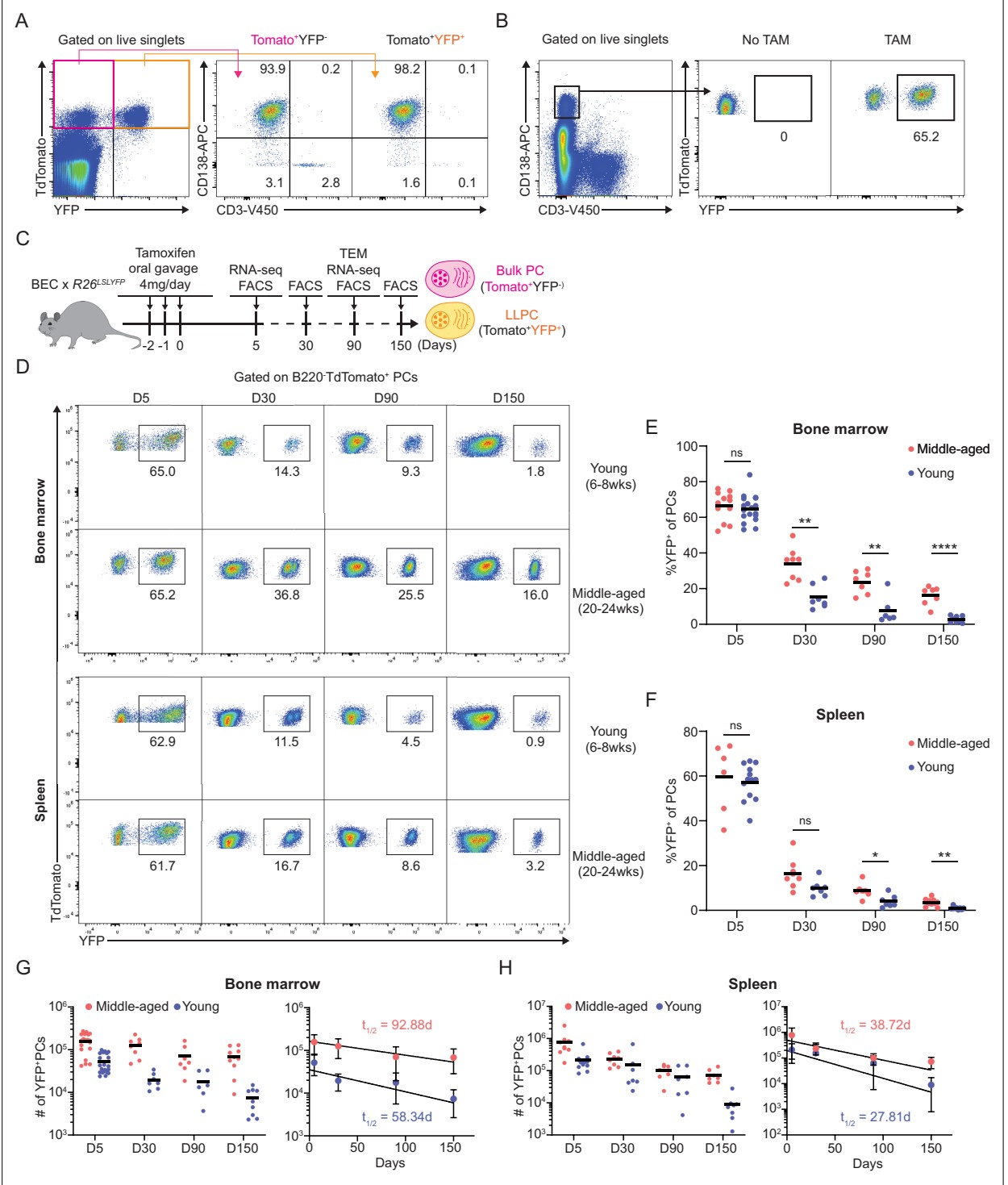

**Figure 1.** Plasma cell (PC) turnover rate decreases with mouse age. (**A**) PC purity of TdTomato+YFP- and TdTomato+YFP+ cells in BEC-YFP mouse at day 5 post tamoxifen treatment. (**B**) Percentage of YFP+ PCs in BEC-YFP mouse in the absence of tamoxifen treatment or treated for 3 consecutive days and analyzed 5 days after the last treatment. (**C**) Experimental setup for measuring homeostatic PC turnover rate in young and middle-aged mice by timestamping at days 5, 30, 90, and 150 after oral gavage tamoxifen treatment, accompanied by transcriptional profiling using bulk RNA-seq at days 5 and 90 and morphological characterization using transmission electron microscopy at day 90. (**D**) FACS pseudo color plots showing decay kinetics of percentage of TdTomato+YFP+ in total B220-TdTomato+ PCs remaining in the bone marrow (BM) (upper panel) and spleen (lower panel) in young and middle-aged mice, quantified in (**E**) for BM and (**F**) for spleen. (**G, H**) Absolute numbers (left panel) and half-lives ($t_{1/2}$) (right panel) of TdTomato+YFP+ PCs in the BM (**H**) and spleen (**I**) in young and middle-aged mice. Curve fitting and $t_{1/2}$ calculations were conducted by using absolute numbers fitted in a

*Figure 1 continued on next page*

**Figure 1 continued**

one-phase decay model. All bars show mean (**E–H**) or mean ± SD (**G, H**). *, p<0.05; **, p<0.01; ****, p<0.0001; ns, non-significant by unpaired Student's t test. All graphs show pooled data from at least two independent experiments (**E**, n=6–18; **F**, n=7–13; **G**, n=6–21; **H**, n=7–13). BEC, Blimp1-ERT2-Cre-TdTomato.

The online version of this article includes the following source data and figure supplement(s) for figure 1:

**Figure supplement 1.** Development of Blimp1-ERT2-Cre-TdTomato (BEC) mice and validation of the fidelity of fluorescent reporter.

**Figure supplement 1—source data 1.** Agarose Gel of PCR product from genomic tail DNA for genotyping mice expressing the BEC allele.

Next we analyzed LLPC spatial organization, as we and others have shown the bulk BM PCs are organized in clusters (*Benet et al., 2021*; *Mokhtari et al., 2015*) and that clusters were sites of reduced PC motility (*Benet et al., 2021*). We used two approaches to determine if LLPCs were more clustered than total bulk PCs. First, we applied our custom script (*Benet et al., 2021*) to identify high-density PC clusters. We masked these regions and found that at late timepoints after tamoxifen, Tomato^bright LLPCs were more enriched in clusters than bulk PCs (*Figure 2F and G*). As this approach can be sensitive to PC densities, we developed a second approach to determine if subsets of PCs were enriched in clusters, based on measuring the nearest distance to 20 PC neighbors (*Figure 2H*). Using this measurement, we found that at day 60, Tomato^bright LLPCs were closer to neighboring PCs (i.e. more clustered) than Tomato^dim bulk PCs, in the BM of both young and middle-aged mice (*Figure 2I*). Taken together, while overall PC motility increases with mouse age, most of the increases in motility can be accounted for by bulk PCs and not by LLPCs, which were relatively sessile. This decrease in LLPC motility is also accompanied by an aggregation or retention in PC clusters, suggesting these are LLPC niches, and may be important for their cell-intrinsic survival or retention in the BM.

## LLPCs exhibit unique surface phenotype and accumulate in the BM with mouse aging

Based on previous datasets (*Lam et al., 2018*; *Akhmetzyanova et al., 2021*; *Shi et al., 2015*; *Cornelis et al., 2020*), we curated a candidate list of 19 PC markers, to identify surface markers of LLPCs. We measured normalized surface expression (fold change) on YFP^+ LLPCs over YFP^- bulk PCs, in the BM and spleen of both young and middle-aged BEC-YFP mice at day 5, 30, 90, or 150 post tamoxifen treatment (*Figure 3A*). We identified six markers that were upregulated (CD93, CD81, CXCR4, and CD326) or two downregulated (CD44 and CD48) with PC age (*Figure 3B*). For the most part, these changes were subtle, whereas CD93 and CD81 expression showed the largest difference in surface expression with PC age. CD93 expression was uniquely bimodal among all tested markers, with YFP^+ LLPCs in BM and spleen were predominantly found in CD93^high subset, in line with genetic evidence for its importance in LLPC maintenance (*Chevrier et al., 2009*; *Figure 3C*). Expression changes of these factors varied based on mouse age and tissue in some but not all cases (*Figure 3—figure supplement 1A–C*), and also varied by isotype depending on the marker (*Figure 3D*), suggesting both intrinsic programs and extrinsic signals control LLPC surface phenotype. Several notable markers important for PC survival were not differentially expressed, including Syndecan-1 (CD138) (*Figure 3—figure supplement 1C*), BCMA (CD269), and TACI (CD267) involved in APRIL signaling (*McCarron et al., 2017*) nor CD28 (*Utley et al., 2020*; *Figure 3A*).

Previous work used glucose uptake, using the fluorescent analog, 2NBDG, as a marker for LLPCs (*Lam et al., 2018*). Indeed, while LLPCs had a high (~80%) frequency of 2NBDG^+ in the BM and spleen (*Figure 3E*), there was no difference between bulk PCs and LLPCs in the BM, indicating that metabolism was not linked with maturation in the BM.

Based on changes in PC turnover with age, we hypothesized the overall BM PC pool may become enriched with LLPCs with aging. Using the six differentially expressed surface receptors, we developed an LLPC enrichment panel to identify quasi-LLPCs in wildtype (WT) mice. Using this gating approach, we could enrich for YFP^+ LLPCs in BEC-YFP mice up to sixfold (*Figure 3F*, *Figure 3—figure supplement 1E*). Using this LLPC panel, we found that middle-aged mice had a higher frequency of quasi-LLPCs within the BM PC compartment as compared to young mice (*Figure 3G*).

Previous transmission electron microscopy (TEM) studies have shown changes in morphology during PC maturation (*Joyner et al., 2022*; *Fooksman et al., 2010*). We also sorted YFP^+ LLPCs and YFP^- bulk PCs from the spleen and BM at day 90 and conducted TEM to see if morphological

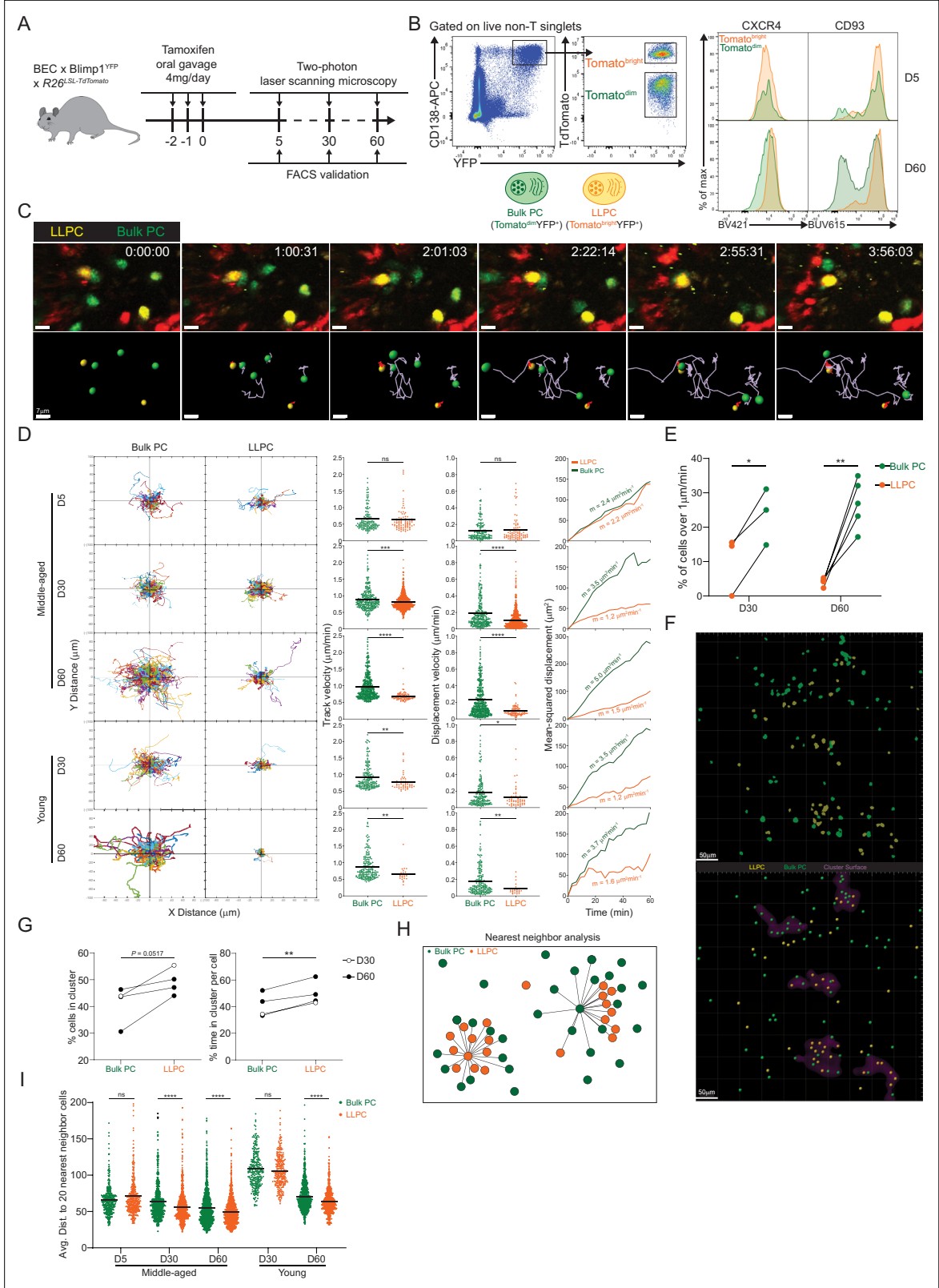

**Figure 2.** Bone marrow (BM) long-lived plasma cells (LLPCs) display cell-intrinsic arrest and clustering. (**A**) Experimental setup for intratibial two-photon intravital imaging of both TdTomato^dimYFP+ bulk plasma cells (PCs) and TdTomato^brightYFP+ LLPCs in the same young or middle-aged mouse by timestamping at days 5, 30, and 60 after oral gavage tamoxifen treatment. (**B**) FACS gating strategy (left panel) for TdTomato^dimYFP+ bulk PCs and TdTomato^brightYFP+ LLPCs after intravital imaging and surface CXCR4 and CD93 expression (right panel) on TdTomato^dimYFP+ bulk PCs compared to

*Figure 2 continued on next page*

*Figure 2 continued*

TdTomato^bright^YFP+ LLPCs at day 5 (control timepoint) and day 60 post tamoxifen treatment. (**C**) Time-lapse images highlighting the cell migration trajectories of four bulk PCs (green spot with light purple tracks) and two LLPCs (yellow spots with red tracks) in a small region of BM parenchyma. Scale bars, 7 µm. (**D**) Individual cell tracks of total bulk PCs or LLPCs plotted at a common origin (left panel) in young and middle-aged mice at days 5, 30, and 60 post tamoxifen treatment. Comparison of total bulk PCs and LLPCs track velocity (middle left panel), track displacement velocity (middle right panel), and mean-squared displacement (right panel) in young and middle-aged mice at days 5, 30, and 60 post tamoxifen treatment. (**E**) Fractions of fast-moving cells (track velocity >1 µm/min) in bulk PCs compared to LLPCs at days 30 and 60 post tamoxifen treatment. Data were pooled from young and middle-aged mice for each timepoint. (**F**) Representative intravital 3D flattened image of masked intensity channels of bulk PCs (green) and LLPCs (yellow) (upper panel) and PC spots (bulk PCs in green and LLPCs in yellow) and cluster surfaces (purple) (lower panel) identified for analysis in (**G**). Scale bars, 50 µm. (**G**) Average percentage of bulk PCs compared to LLPCs staying inside cluster surface over time (left panel). Average time of individual bulk PCs spent inside cluster surface compared to that of LLPCs (right panel). (**H**) Depiction of nearest neighbor analysis for cell-cell distance (lines) in 3D (collapsed in 2D) between bulk PCs (green) and total PCs (green and orange), or between LLPCs (orange) and total PCs (green and orange), which is quantified in (**I**) for an average distance between bulk PCs or LLPCs and their 20 nearest neighbor cells (total PCs combining bulk PCs and LLPCs). Each symbol represents one randomly picked cell per subset, and data were pooled from at least two mice. All bars show mean (**D, I**). *, p<0.05; **, p<0.01; ***, p<0.001; ****, p<0.0001; exact p-values; ns, non-significant by Mann-Whitney U test (**D**), Kruskal-Wallis test with Dunn's test for multiple comparisons (**I**), or paired Student's t test (**E, G**). All graphs show pooled data from at least two independent experiments (**E**, n=3–5; **G**, n=4).

The online version of this article includes the following video for figure 2:

**Figure 2—video 1.** Maturation-dependent long-lived plasma cell (LLPC) motility by two-photon intravital imaging in the bone marrow.

https://elifesciences.org/articles/89712/figures#fig2video1

differences accompanied LLPC maturation (*Figure 3—figure supplement 2*). Overall, we did not detect statistically significant differences in cell size, cytoplasmic area, mitochondrial density between mature PC subsets, although the distributions had wide ranges. There were minor yet significant changes in nuclear size and chromatin density in splenic LLPCs over BM LLPCs. Taken together, we conclude that differential surface protein expression accompanies cell-intrinsic LLPC maturation, but otherwise cells appear morphologically similar.

## CXCR4 controls durability of humoral responses by promoting PC survival and retention in the BM

CXCR4 is the master chemokine receptor required for lymphocyte entry and retention in the BM (*Zehentmeier and Pereira, 2019*). Based on its important role in BM PC motility and retention (*Benet et al., 2021*), and its upregulated expression on LLPCs (*Figure 3B and D*), we decided to test if it is required in PCs specifically to maintain humoral responses. Previous work (*Nie et al., 2004*) had shown that conditional deletion of *Cxcr4* using a pan B cell expressing cre (CD19-cre) was dispensable for humoral responses and PC survival following vaccination, but potentially this approach did not specifically target PCs and may not be fully penetrant (*Aaron and Fooksman, 2022*).

We bred BEC *Rosa26^LSLYFP^ Cxcr4^fl/fl^* mice (or CXCR4^cKO^), which would result in CXCR4 deletion in PCs upon TAM treatment. Cohorts of control BEC-YFP (here referred to as WT) and CXCR4^cKO^ mice were immunized (on day –30) with NP-KLH/alum to generate similar NP-specific PCs and titers at day –3 (*Figure 4A and B*, *Figure 4—figure supplement 1A–C*), at which point, they received tamoxifen to induce *Cxcr4* deletion in CXCR4^cKO^ mice and fate-label PCs with YFP in both groups of mice. We confirmed that *Cxcr4* expression was diminished specifically in YFP+ PCs in CXCR4^cKO^ mice at mRNA transcript and protein levels (*Figure 4C*, *Figure 4—figure supplement 1A*) at day 60 in the BM and splenic LLPCs but not in control bulk PCs. Interestingly, CXCR4 surface protein levels were significantly reduced but not completely lost, suggesting incomplete deletion in some cells. Nevertheless, anti-NP titers declined faster in CXCR4^cKO^ mice as compared to WT controls (*Figure 4B*). Decreases in anti-NP titers were associated with reduced numbers of NP-specific LLPCs (YFP+) in spleen and BM of CXCR4^cKO^ mice as compared to WT mice (*Figure 4D*).

To determine the role of CXCR4 in homeostatic PC turnover and LLPC competition, we generated chimeric animals using 1:1 ratio of congenically labeled cells from WT and CXCR4^cKO^ mice. For these studies, mice expressing BEC *Rosa26^LSL-Tomato^* alleles were used as WT controls. Eight weeks post reconstitution (*Figure 4E*), mice were treated with tamoxifen and PC decays were tracked over 90 days by flow cytometry. At day 5, fewer PCs were found in the CXCR4^cKO^ vs WT compartment in the BM and spleen (*Figure 4F*, *Figure 4—figure supplement 1D*), suggesting labeling efficiency was reduced or there was rapid decline in KO PCs cells from the tissue. Correcting for their relative abundance at

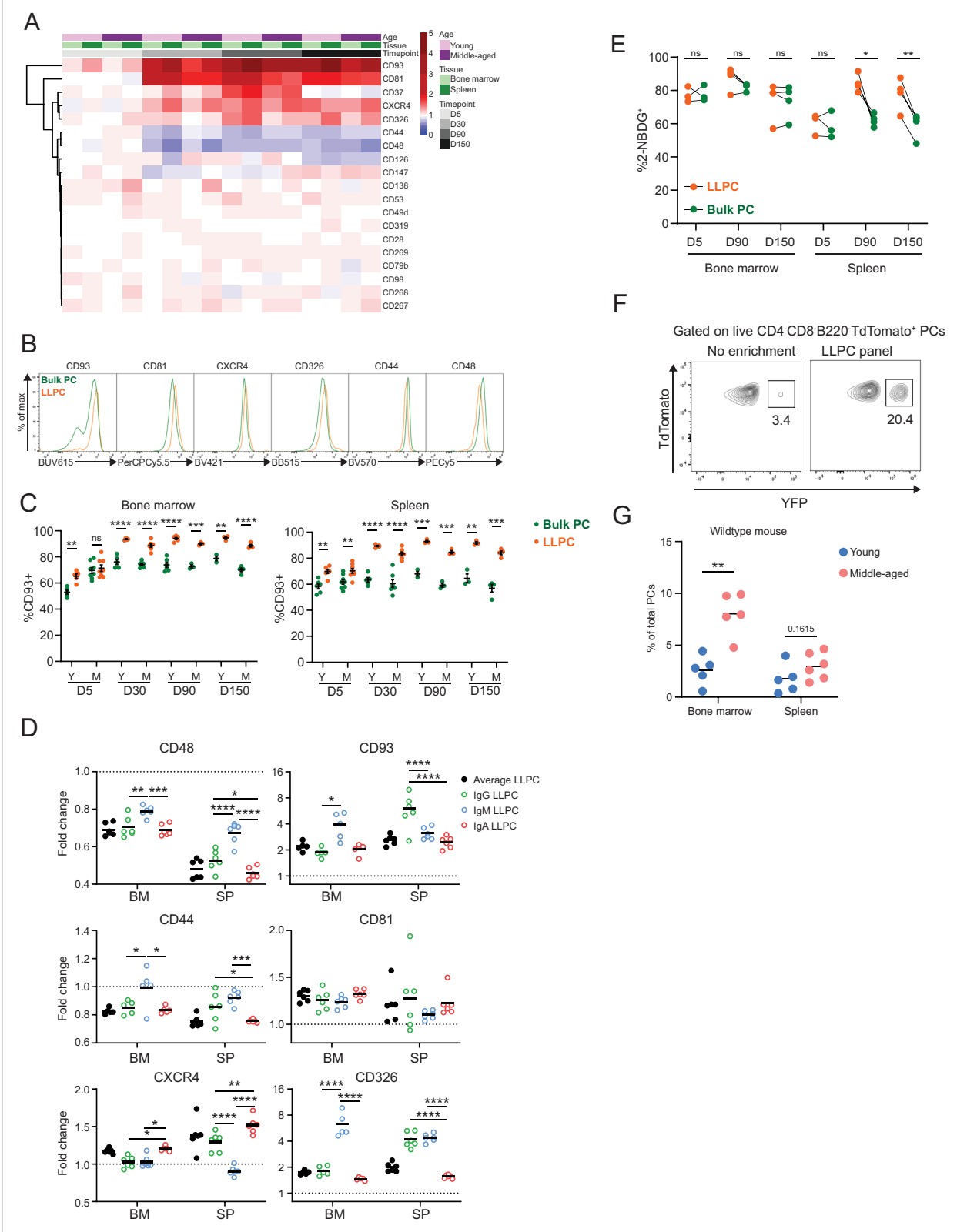

**Figure 3.** Differentially expressed surface receptors accompany long-lived plasma cell (LLPC) maturation. (**A**) Heatmap depicting average fold changes of surface marker expression level (gMFI) on TdTomato⁺YFP⁺ LLPCs and TdTomato⁺YFP⁻ bulk plasma cells (PCs) from three to four mice per unit at indicated timepoints post tamoxifen treatment in the bone marrow and spleen of young and middle-aged mice. Color scale showing fold increase in red, fold decrease in blue, or no difference in white. (**B**) Overlay histograms comparing the expression level of differentially expressed surface markers

*Figure 3 continued on next page*

*Figure 3 continued*

on TdTomato⁺YFP⁻ bulk PCs to TdTomato⁺YFP⁺ LLPCs at day 90 post tamoxifen treatment. (**C**) Percentage of CD93⁺ cells on TdTomato⁺YFP⁺ LLPCs relative to TdTomato⁺YFP⁻ bulk PCs in individual mouse at indicated timepoints post tamoxifen treatment in the bone marrow (left panel) and spleen (right panel) of young and middle-aged mice. (**D**) The fold change of the expression level (gMFI) of differentially expressed surface markers of isotype-specific or total TdTomato⁺YFP⁺ LLPCs relative to matching isotype TdTomato⁺YFP⁻ bulk PCs in individual mouse at day 90 post tamoxifen treatment in young BEC-YFP mice. (**E**) Analysis of 2NBDG⁺ population within LLPC and bulk PC subsets at indicated timepoints. (**F**) Percentage of TdTomato⁺YFP⁺ LLPCs pre and post enrichment using a combination of six antibody panel of differentially expressed surface markers identified in (**B**). No enrichment represents the percentage of TdTomato⁺YFP⁺ LLPCs at day 90 post tamoxifen in young mice. (**G**) Percentage of quasi-LLPCs in total PCs identified using six-marker antibody panel in young and middle-aged mice at steady state. All bars show mean ± SEM (**C–E**). *, p<0.05; **, p<0.01; ***, p<0.001; ****, p<0.0001; ns, non-significant by unpaired Student's t test. All graphs show pooled data from at least two independent experiments (**A, C–E**, n=3–7; **G**, n=5–6 mice). BEC, Blimp1-ERT2-Cre-TdTomato.

The online version of this article includes the following figure supplement(s) for figure 3:

**Figure supplement 1.** Long-lived plasma cell (LLPC) enrichment by a multiplexed antibody panel.

**Figure supplement 2.** Transmission electron microscopy of long-lived plasma cells (LLPCs).

day 5, labeled WT (Tomato^bright) PCs in BM hardly decayed over 90 days, whereas ~50% of CXCR4^cKO were lost (*Figure 4G*, *Figure 4—figure supplement 1E*). Within the spleen, PC turnover was overall more rapid, with a 50% and 90% loss of WT and CXCR4^cKO labeled PCs, respectively. Overall, WT PCs outcompeted CXCR4^cKO PCs in the BM and spleen over time as assessed by competency ratio (*Figure 4H*, *Figure 4—figure supplement 1F*). We analyzed changes in key PC pro-survival factors, Mcl1 and Bcl2 (*Figure 4I and J*), and found that WT labeled PCs had higher relative expression than CXCR4^cKO counterparts, and while they had similar levels at day 5, PC survival was compromised by loss of CXCR4 over time in bone and spleen suggesting CXCR4 is important for long-term survival of PCs.

CXCR4 signaling can directly promote cell survival via AKT pathway (*Scotton et al., 2002*), but it may act indirectly on PC survival by dislodging from survival niches. Thus, we asked if loss of antigen-specific CXCR4^cKO PCs was due to cell death in the BM, or egress from the BM niche, eventually leading to PC loss. Chimeric mice were intra-tibially (IT) injected with 4-hydroxy-tamoxifen (4-OH-TAM), to induce cre recombination in PC subsets in one bone (*Figure 5A*). We used this approach previously to track recirculation of BM PCs (*Benet et al., 2021*). At day 1 post injection, WT PCs within the injected tibia were the predominant location of labeled PCs, consistent with a local administration and activity (*Figure 5B and C*). However, within CXCR4^cKO PC pool, most of the labeled PCs were predominantly found in the spleen, but also found at higher frequencies in other bones. This subset-specific effect is unlikely due to leakage of 4-OH-TAM to other tissues, as it would have affected both groups of PCs equally. Thus, the likely conclusion is that CXCR4^cKO must have rapidly egressed the BM upon cre-deletion of *Cxcr4*. Over time labeled WT PCs egressed the tibia and redistributed to other sites whereas the labeled CXCR4^cKO PC remained fixed in spleen and other niches (*Figure 5D*). To confirm this effect was due to rapid egress, mice were pretreated with pertussis toxin (PTX), which we had found could block PC motility in the BM (*Benet et al., 2021*). Pretreatment with PTX prevented CXCR4^cKO PCs from accumulating in the spleen, following IT administration of 4-OH-TAM (*Figure 5E*). Thus, deletion of *Cxcr4* triggers rapid mobilization of PCs from the BM, suggesting dislodging PCs from their niche occurs prior to defects in cell survival.

## Shared transcriptional program accompanies BM and splenic LLPC specification

As ASCs mature and migrate to the BM, their transcriptome changes (*Shi et al., 2015*). Based on the changes in surface expression, we hypothesized that LLPCs may also encode a unique transcriptome that fuel these protein expression differences. Based on previous studies of PC transcriptome studies (*Lam et al., 2018*), we expected mRNA expression differences in LLPCs to be minor, and due to the over-representation of immunoglobulins in the transcriptome, we performed bulk RNA-seq with deep reads (50 million reads per sample) to improve our resolution of global changes. For these studies, we FACS-purified matching populations of YFP⁺ LLPCs and YFP⁻ bulk PCs from BM and spleen of BEC-YFP mice, on day 90 post tamoxifen treatment. We used groups from both young and middle-aged mice for these studies to see what effect mouse age played in gene expression or PC composition. As negative controls, we also sorted YFP⁺ and YFP⁻ PCs from middle-aged mice, on day 5 post treatment.

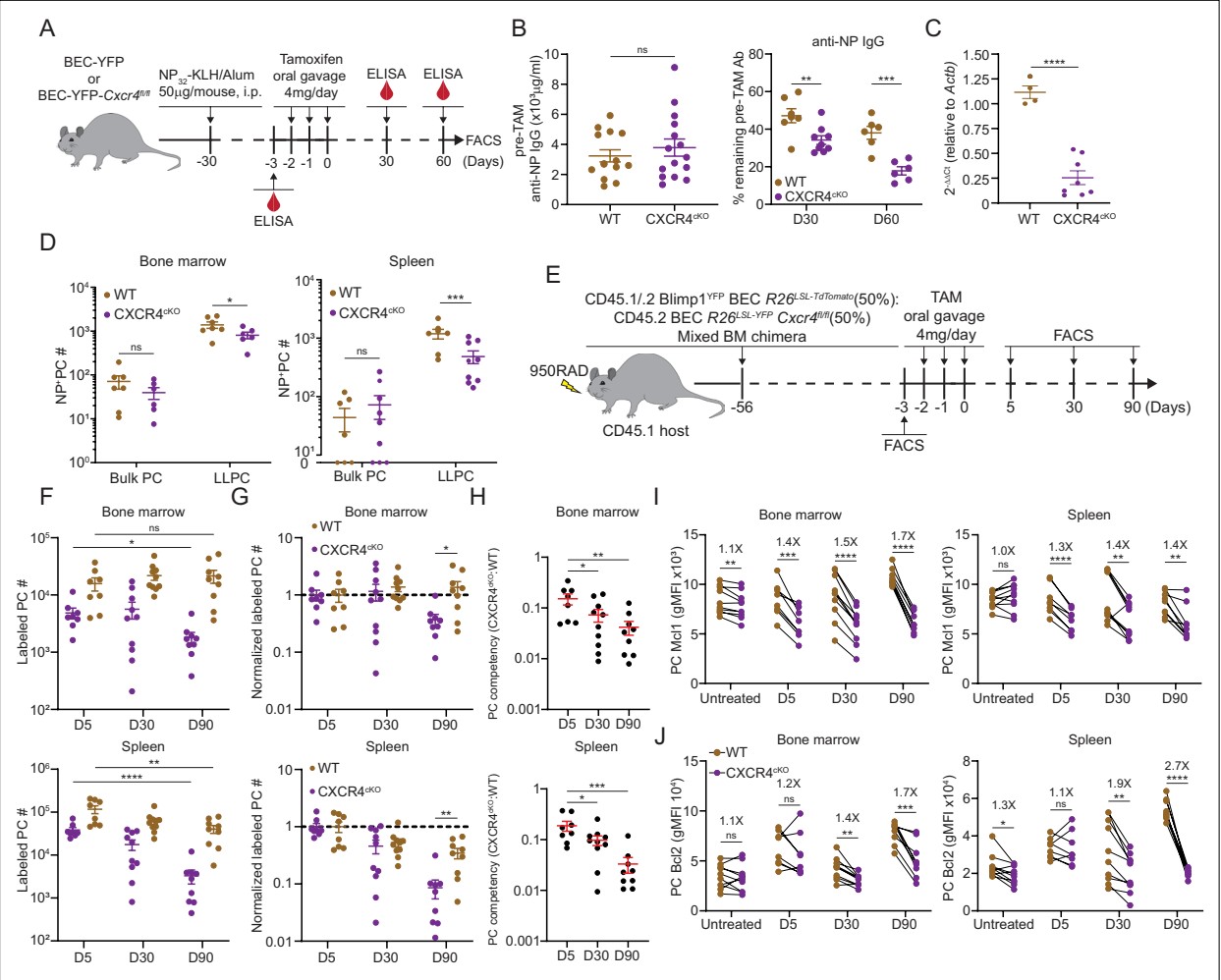

**Figure 4.** CXCR4 controls durable humoral response by promoting plasma cell (PC) survival. (**A**) Experimental setup for examining the role of CXCR4 in sustaining antigen-specific antibody responses using an NP-KLH/alum immunization model. i.p., intraperitoneal. (**B**) Anti-NP antibody titer in wildtype (WT) or CXCR4$^{cKO}$ mice before tamoxifen treatment at day 30 post immunization (left panel) and the remaining percentage of pretreat anti-NP antibody at indicated timepoints post tamoxifen treatment (right panel). (**C**) Fold change of FACS-purified YFP$^+$ PCs $Cxcr4$ mRNA level in WT or CXCR4$^{cKO}$ mice, normalized to $Actb$ levels in WT mice. (**D**) Absolute numbers of NP-specific bulk PCs and long-lived plasma cells (LLPCs) in the bone marrow (BM) (left panel) and spleen (right panel) of WT and CXCR4$^{cKO}$ mice. (**E**) Experimental setup for generating mixed BM chimera reconstituted with WT and CXCR4$^{cKO}$ donors. (**F**) Absolute number of (YFP$^+$) labeled BM and spleen PCs at indicated timepoints post tamoxifen treatment (**G**) normalized labeled PC numbers shown in (**F**) (relative to day 5 post tamoxifen treatment). (**H**) PC competitive competency (right panel) at indicated timepoints determined by normalizing the CXCR4$^{cKO}$:WT ratio in the BM labeled PC compartment to that of total splenic B cell compartment (upper panels) or the CXCR4$^{cKO}$:WT ratio in the splenic labeled PC compartment to that of total splenic B cell compartment (lower panels). (**I, J**) Mcl1 (**I**) and Bcl2 (**J**) intracytoplasmic expression (by gMFI) of WT or CXCR4$^{cKO}$ labeled PC compartment at indicated timepoints in the BM (left panel) and spleen (right panel). Fold changes of WT over CXCR4$^{cKO}$ labeled PCs in Mcl1 and Bcl2 expression level are indicated above the statistical significance symbol. All bars show mean ± SEM. Each symbol in all plots represents one mouse. *, p<0.05; **, p<0.01; ***, p<0.001; ****, p<0.0001; ns, non-significant by unpaired Student's t test (**B, C, D, F–G, L, and M**), paired Student's t test (**I, J**) or one-way ANOVA with multiple comparison correction using the Holm-Šídák test (**H**). All graphs show pooled data from two independent experiments (**B**, n=13–15 (left), n=6–9 (right); **C**, n=4–8; **D**, n=6–9; **F–H**, n=8–10; **I**, n=8–10; **J**, n=8–10).

The online version of this article includes the following figure supplement(s) for figure 4:

**Figure supplement 1.** CXCR4 controls durable humoral response by promoting plasma cell (PC) survival and retention in the bone marrow (BM).

In all, we analyzed 12 groups of PCs (n=3/4 per group, 44 samples total). For these global analyses, we excluded immunoglobulin genes, but analyzed them separately in the next section.

We performed unsupervised clustering of day 90 PC samples using all differentially expressed genes (DEGs) measured by pair-wise comparisons of YFP$^+$ and YFP$^-$ samples, from matching tissues (p$_{adj}$<0.05, with no cut-offs for fold change or reads). Based on the sample dendrogram, we found

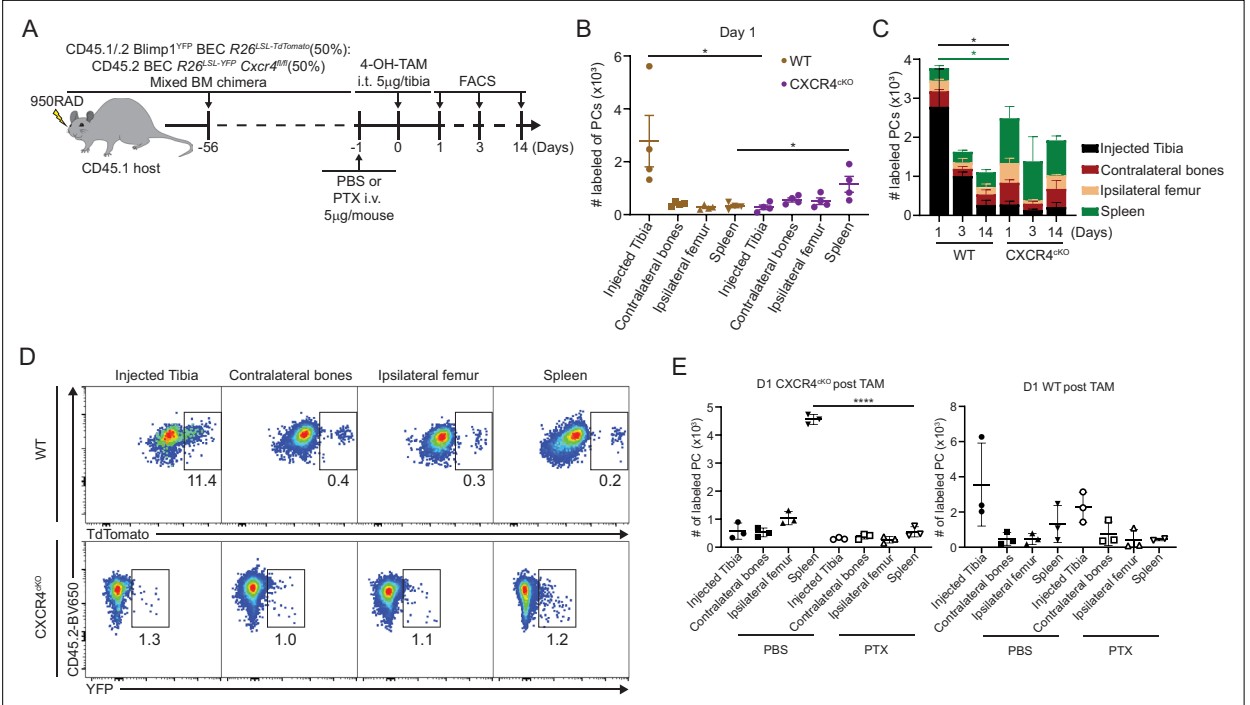

**Figure 5.** CXCR4 retains plasma cells (PCs) in the bone marrow (BM). (**A**) Experimental setup for intratibial injection into WT:CXCR4$^{cKO}$ mixed BM chimera in the presence or absence of pertussis toxin (PTX) treatment. (**B**) Absolute number of labeled WT or CXCR4$^{cKO}$ PC in injected tibia and distal organs/tissues at day 1 post 4-hydroxy-tamoxifen (4-OH-TAM) injection. (**C**) Distribution of absolute numbers of labeled WT or CXCR4$^{cKO}$ PCs in injected tibia and distal organs/tissues. (**D**) FACS pseudo color plot showing the percentage of labeled PCs in injected tibia and other distal organs/tissues at day 1 post intratibial injection. (**E**) Absolute number of labeled CXCR4$^{cKO}$ (left panel) or WT (right panel) PCs in injected tibia and distal organs/tissues at day 1 post 4-OH-TAM injection in the presence or absence of PTX. All bars show mean ± SEM. Each symbol in all plots represents one mouse. *, p<0.05; ****, p<0.0001; ns, non-significant by unpaired Student's t test (**E**), paired Student's t test (**B**), or one-way ANOVA with multiple comparison correction using the Holm-Šídák test (**E**). All graphs show pooled data from two independent experiments (**B**, n=4; **E**, n=3).

that most LLPC (yellow) and bulk PC (red) samples clustered separately, and within the LLPCs, BM (light green) and splenic (dark green) samples were closely related (column headings on *Figure 6A*). Unsupervised clustering of DEGs (rows) revealed five groups of genes (*Figure 6A*, *Figure 6—source data 1*). Specifically, groups 5 and 2 genes contained DEGs that were either upregulated or downregulated in all LLPC groups, respectively. Group 1 genes were specifically upregulated in splenic LLPCs, suggesting tissue-specific expression patterns. To better understand the overlaps or similarities of these LLPC groups, we generated an UpSet analysis plot (*Conway et al., 2017*; *Lex et al., 2014*), based on pair-wise comparisons (*Figure 6B*). Splenic LLPCs from young and middle-aged mice had the most DEGs, likely because splenic (YFP⁻) bulk PCs used in comparisons were highly enriched in short-lived PCs. Among DEGs shared among LLPC subsets, many were shared by three or four of groups. 12 DEGs were shared by all LLPCs (*Cd55*, *Cxcr3*, *Cyp4f18*, *Fam3c*, *Gpx3*, *H2-Aa*, *H2-Ab1*, *Hcst*, *Prss57*, *Rab3b*, *Slamf6*, *Spag5*). As negative controls, comparisons of day 5 YFP⁺ vs YFP⁻ PCs were conducted, and as expected, very few DEGs were detected or shared with LLPCs. Using circle plots (*Figure 6C*), we summarized the overlaps of DEGs, and found that BM LLPCs DEGs were more commonly shared among LLPC groups, as compared to splenic LLPCs. We also observed that LLPCs from middle-aged mice had fewer DEGs than young mice, consistent with the view that bulk PCs (YFP⁻) are enriched with LLPCs in older mice (*Figure 4C*).

Next, to determine which biological pathways were altered in LLPCs, we generated gene ontology (GO) terms based on the previously identified DEGs, and assessed term enrichment in LLPC subsets and day 5 control groups (*Figure 6D*). As expected, LLPCs showed downregulation of MHC Class II pathway and proliferation-related pathways. In contrast, LLPCs showed increased cell survival and stress response pathways, increased lipid metabolism, and neural-immune signaling. Changes in cell adhesion and chemotaxis were highly enriched in LLPCs and there were also changes in cytokine production pathways. From the total DEG list, putative cell surface receptors were extracted to

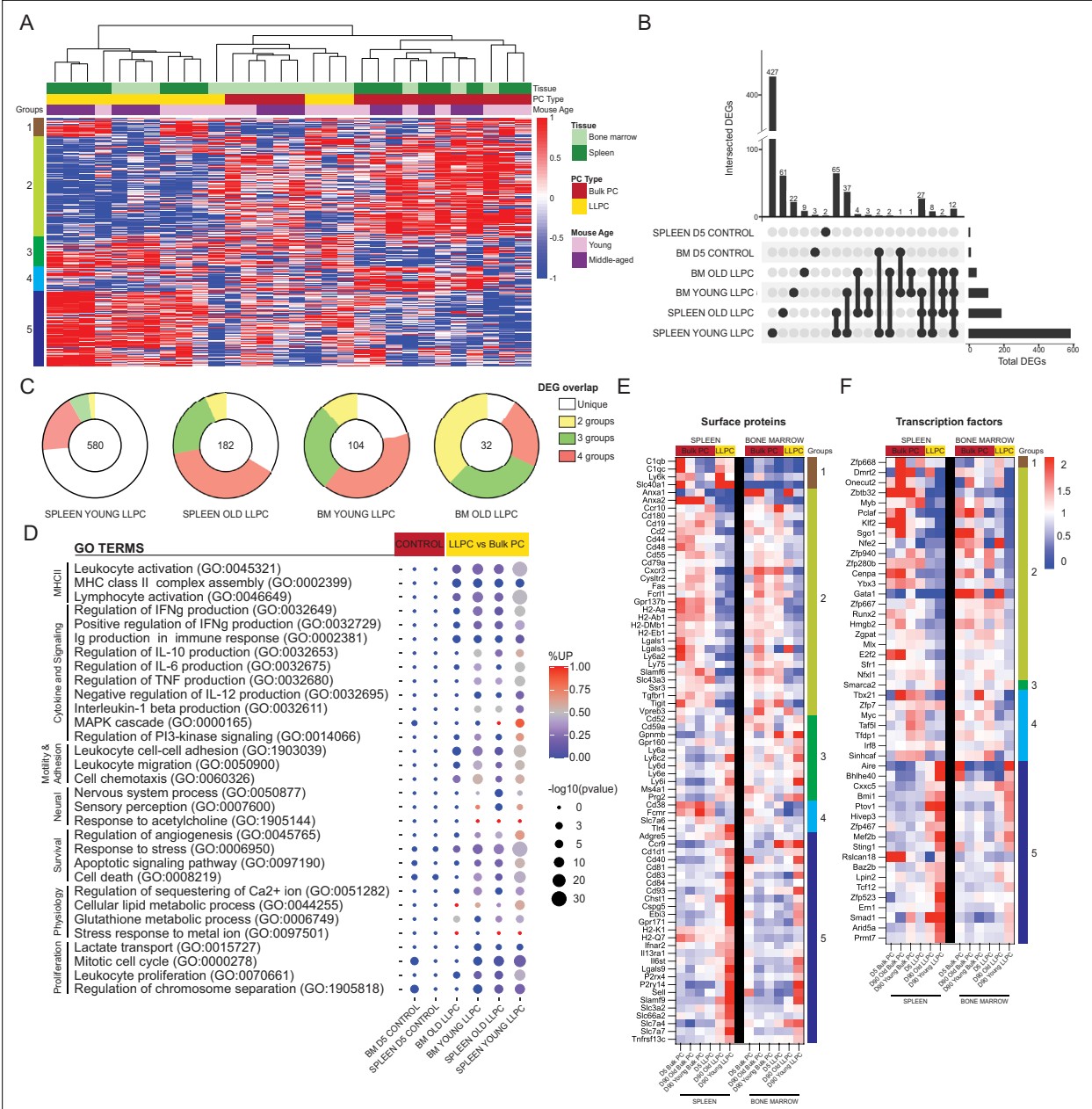

**Figure 6.** Shared transcriptional program accompanies bone marrow (BM) and splenic long-lived plasma cell (LLPC) specification. (**A**) Heatmap depicting unsupervised clustering of total differentially expressed genes (DEGs, p_adj-value<0.05) between TdTomato⁺YFP⁻ bulk plasma cells (PCs) to TdTomato⁺YFP⁺ LLPCs across tissue types (bone marrow and spleen) and mouse ages (young and middle-aged) at day 90 post tamoxifen treatment, with no cut-off for fold change and transcripts per million reads (TPM). Color scale represents z-score for normalization per gene (row). Total DEGs were separated in five color-coded clusters. (**B**) UpSet plot visualizing total number of DEGs in each pair-wise comparison (single node) and intersections (connecting nodes) between DEGs among different pair-wise comparisons. (**C**) Pie charts showing the fractions of the DEGs that are unique in one pair-wise comparison group or shared by two to four groups of pair-wise comparisons. Numbers in the center of each chart represent the total number of DEGs in each indicated pair-wise comparison group. (**D**) Bubble plots showing selected gene ontology terms (GO terms) enrichment comparing LLPCs groups (highlighted in yellow) and day 5 control groups bulk PCs (highlighted in red) based on previously identified DEGs in (**A**). Color scale bar showing the percentage of DEGs upregulated per GO term in each pair-wise comparison group (red, >50% upregulated DEGs; blue, <50% upregulated DEGs). Circle size represents the significance of the enrichment based on the -log10(p-value). (**E, F**) Heatmap of all DEGs encoding surface proteins (**E**) or transcription factors (**F**) between LLPC groups and bulk PC groups in both the spleen (left) and the bone marrow (right), which are further separated by gene clusters identified in (**A**).

The online version of this article includes the following source data for figure 6:

**Source data 1.** Excel spreadsheet listing differentially expressed genes (DEGs) and cluster heading shown in *Figure 6A*.

generate heatmaps of normalized expression among PC subsets in the BM and spleen, clustered by DEG groups (*Figure 6E*). These included chemokine receptors (*Ccr10*, *Cxcr3*, *Ccr9*, *S1pr1*, *Ebi3*), adhesion molecules (L-selectin, *Ly6* family, Galectins, *Cd93*), MHC-related molecules, co-stimulatory factors (SLAM family, *Tigit*), and cytokine receptors (*Il6st*, *Il13ra1*, *Ifnar2*, *Tgfb2*) to name a few. Some of these LLPC factors were tissue-specific, such as *C1q* and *Adgre5* expressed by splenic LLPCs. There were notable absences from the list, such as *Cxcr4*, which is upregulated at the protein level in LLPCs, suggesting that minor changes in transcripts may be regulating larger changes at the protein level, or important regulation may be occurring post-transcriptionally (*Greenbaum et al., 2003*).

We also generated a putative list of transcription factors (TFs) and chromatin-remodeling factors differentially expressed in LLPCs (*Figure 6F*). Among known PC-related factors, *Bmi1* (*Di Pietro et al., 2022*) was upregulated in LLPCs while *Myb* (*Good-Jacobson et al., 2015*), *Klf2* (*Winkelmann et al., 2011*), and *Zbtb32* (*Jash et al., 2019*) were downregulated. Interestingly, *Aire* (*Mathis and Benoist, 2009*) was among the most upregulated LLPC genes, but its role in PCs has not been explored. Many classical PC TFs were not differentially expressed by LLPCs, including *Prdm1* (encoding Blimp1), *Irf4*, and *Xbp1*. Taken together, murine LLPCs exhibit a unique global transcriptome, fine-tuning surface receptors, and transcriptional factor expression, which may support longevity.

## LLPC receptors have reduced BCR diversity but enriched in public clones

As expected, the major RNA transcript in these PCs were immunoglobulin heavy and light chains. We assembled over 26,000 complete clones (but not paired sequences) for the BCR heavy and light loci and analyzed their clonal properties, to determine if LLPCs had unique features in different tissues and are from mice of different ages. Notably, these are unselected polyclonal PCs from naïve mice, with unknown antigen specificities.

First, we analyzed isotype usage and found that IgA PCs were the major isotype within the BM and also in the spleen, to a lesser extent (*Figure 7A*). The one notable exception was that splenic LLPCs from young mice that were timestamped at 6–8 weeks of age were highly enriched in IgM, in comparison to all other samples, including splenic bulk PCs from same (young) mice. This suggest that these splenic IgM LLPCs are specified early in life and tend to be selectively retained, and maybe derived from B-1 lineages (*Baumgarth, 2016*). Within LLPCs subsets, both in young and middle-aged mice, BM LLPCs have a higher IgA:IgM composition as compared to splenic LLPC counterparts, suggesting tissue-specific homing or retention of different LLPCs on the basis of isotype.

Next, we analyzed diversity of clones in the LLPC and bulk PC subsets, by comparing the heavy chain V-segment+CDR3 exact amino acid sequences. We found that LLPC samples (day 90 YFP[+] BM and spleen) had reduced clonal diversity based on Chao1 estimation index (*Figure 7B*) as compared to bulk (day 90 YFP[-] BM and spleen) PCs, while no differences were observed between YFP[+] and YFP[-] subsets at day 5 post TAM. This suggested LLPCs had a reduced repertoire and complexity compared to bulk PCs. This also raised the possibility that different subsets of clones were selected for LLPC fate specification.

To see if LLPC and bulk PCs arise from same pool of B cells, we calculated frequencies of shared clones from different PC groups within the same mouse (*Figure 7C*). As expected, PCs from day 5-treated mice showed the highest overlap of clones between YFP[+] and YFP[-] subsets in all tissues, along with bulk (YFP[-]) PCs in BM and spleen, consistent with their heterogenous PC phenotypes and coordinated timing for PC differentiation. LLPCs in the BM and spleen had more clonal overlap than with matched bulk PCs taken from the same sites. This trend was more striking in young mice, which have fewer LLPCs in the YFP[-] subset than in middle-aged mice, suggesting these LLPCs may arise from a different clonal population of B cells in young mice than in middle-aged mice.

LLPCs have been suggested to arise from germinal centers, which could suggest that residency time in the GC may regulate selection into the LLPC pool. To see if polyclonal LLPCs were more highly mutated than total bulk PCs, we compared overall somatic hypermutation (SHM) frequencies in V regions matching samples of YFP[+] (LLPC) and YFP[-] (bulk PCs) and found that day 90 LLPC clones had fewer mutations than bulk PCs in BM or spleen whereas no differences in mutations were observed between day 5 samples (*Figure 7D*). When averaging all clones per sample, LLPCs also had fewer SHM than paired bulk PC samples in the same tissue of the same mouse (*Figure 7E*). LLPCs timestamped

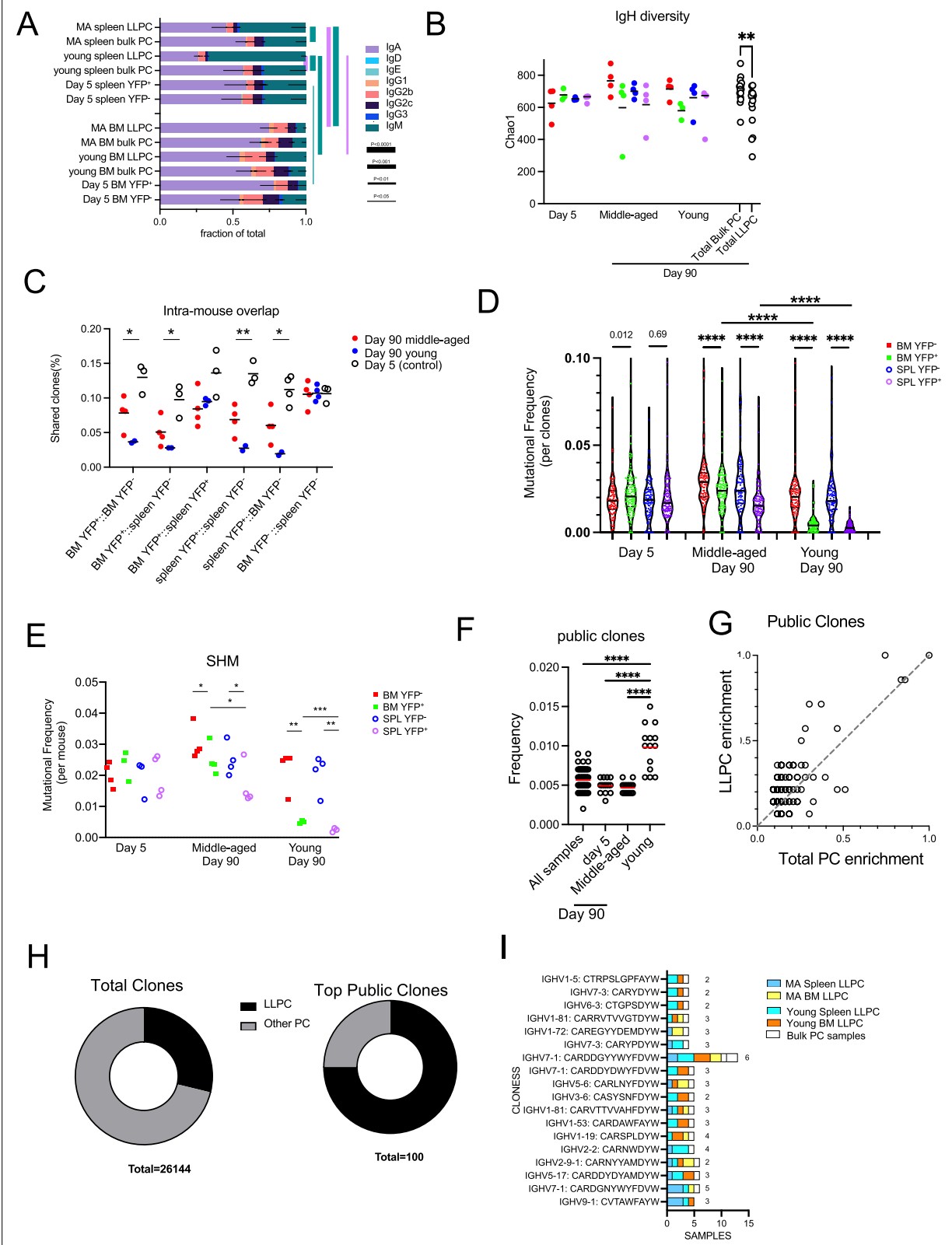

**Figure 7.** Long-lived plasma cell (LLPC) receptors have reduced BCR diversity but enriched in public clones. (**A**) Stacked bar plots showing isotype gene usage (the fraction of isotypes in total mapped complete clones per group) in LLPCs and bulk plasma cells (PCs) at days 5 and 90 post tamoxifen treatment across tissue types (spleen and bone marrow [BM]) and mouse ages (young and middle-aged). Line thickness represents the statistical significance based on p-value thresholds. Line color represents the comparisons of color-coded isotypes between groups. (**B**) BCR repertoire diversity

*Figure 7 continued on next page*

*Figure 7 continued*

of LLPCs and bulk PCs at indicated timepoints across tissue types (spleen and BM) and mouse ages (young and middle-aged), estimated by Chao1 richness index for the abundance of unique clones in a repertoire per group. Each symbol represents one mouse, and pooled bulk PC samples comparing pooled LLPC samples at day 90 post tamoxifen treatment were shown on the right in black. (**C**) The percentage of clones shared by indicated PC subsets within the same mouse (intra-mouse). X axis format (**A, B**) reflects samples A and B used for comparison. (**D**) Violin plots comparing the distribution of somatic mutation frequencies per specific V region (across all clones) in YFP⁻ bulk PCs and YFP⁺ LLPCs at indicated timepoints in young and middle-aged mice. Each symbol represents one clone, and multiple samples were pooled from each group. (**E**) Average mutation frequencies of all clones in (**D**) for each PC subset in indicated tissues in young and middle-aged mice. Each symbol represents one mouse. (**F**) The percentage of clones shared by all samples, day 5 samples, middle-aged mice samples, and young samples. Each symbol represents one sample (e.g. BM YFP⁺ sample, BM YFP⁻ sample, etc.). (**G**) Frequency of top 100 most abundant public clones in all PC samples compared to that in LLPC samples. Clones showing no preference for LLPC samples over total samples are distributed on the diagonal line in the plot. Each symbol represents a clone. (**H**) Pie charts showing the fraction of total clones or top 100 most abundant public clones in LLPCs compared to other PCs. (**I**) Stacked bar plots showing the distribution of sample types (in tissues and age of mice) per most frequent public clone in LLPCs. Number on the right of each stacked bar represents the number of mice. *, p<0.05; **, p<0.01; ***, p<0.001; ****, p<0.0001; exact p-values for non-significance by unpaired Student's t test.

The online version of this article includes the following figure supplement(s) for figure 7:

**Figure supplement 1.** Analysis of public clones in plasma cell (PC) samples.

at 6–8 weeks (young mice) had even fewer mutations than LLPCs in middle-aged mice. Overall, this confirms that PC cells need not arise from affinity-selected GC B cells in order to enter the LLPC pool.

Finally, to see if certain clones were 'public' (or shared by at least two samples from different mice), we analyzed heavy chain clonal overlap (as in *Figure 7B and C*) in all samples. Young mice had the highest overlap of shared public clones compared to other groups (*Figure 7F*). We analyzed which groups of PCs were responsible for this elevation and found that LLPCs in BM and spleen accounted for most of the public clones (*Figure 7—figure supplement 1A*). Next, we analyzed the top 100 most abundant public clones across all samples (*Figure 7—figure supplement 1B*) and found a biased enrichment toward LLPC samples (*Figure 7G*). While LLPCs represented about 28% of all found clones (n=26,144), in the top public clones across samples, 75% were found in LLPC samples from multiple tissues and mice (*Figure 7H*). Among the top LLPC public clones (found in >75% of LLPC samples, *Figure 7—figure supplement 1B*), they were surprisingly absent in bulk PC groups (*Figure 7I*). This suggests that the some of LLPC endogenous repertoire is directly selected into the LLPC compartment for long-term maintenance.

## Discussion

The mechanisms and conditions underlying cell fate into LLPCs following vaccination remains a long-standing question for durable humoral memory. Moreover, once LLPCs are specified, the intrinsic programming and extrinsic factors that control their longevity are still undefined (*Robinson et al., 2020*; *Robinson et al., 2022*). While the field has leaned toward a model that LLPCs arise from late GC B cells (*Weisel et al., 2016*), LLPCs can be generated by T-independent fashion (*Bortnick et al., 2012*) as well. Recent work using similar PC timestamping tools have demonstrated that NP-specific LLPCs can arise from pre-GC stages and accumulate at constant click during the immune response, showing no bias toward late stages (*Robinson et al., 2022*) nor requiring high affinity for longevity, at least in the NP immunization model. In 'naïve' mice, which are not biased by immunization, we find clones selected into LLPCs pool bear fewer somatic mutations than bulk PCs, consistent with recent scRNA-seq analysis (*Liu et al., 2022*), and aforementioned findings that LLPC generation is not strictly dependent on late GC B cells following immunization (*Robinson et al., 2022*; *Koike et al., 2023*). We also find limited diversity in BCR repertoire, which may reflect unique clones or antigens are pre-programmed for LLPC or, merely that longer immune responses engender more clonal LLPC over time (*Robinson et al., 2022*). We also find LLPCs are enriched in public or shared clones, which have been recently shown to be microbial and self-reactive (*Liu et al., 2022*; *Blanc et al., 2016*; *Lino et al., 2018*; *Racine et al., 2011*). Interestingly, while some of our public clone lists have shared V regions with known self-reactive and microbial specificities, many are not found on any of these lists, suggesting variations in microbiome composition or diet may shift the LLPC clonal composition.

Within the BM, we found that IgA⁺ LLPCs are the major subset, similar to the bulk BM PC composition, in line with previous studies (*Xu et al., 2020*; *Liu et al., 2022*), and likely depend on microbial composition in the gut (*Liu et al., 2022*; *Wilmore et al., 2018*). In contrast, the majority of splenic

LLPCs are a mix of IgM+ and IgA+ LLPCs, and particularly in young mice, IgM+ LLPCs seem to be specified early and selectively retained in the splenic niche. While IgM+ LLPCs can be generated by various pathways (*Blanc et al., 2016*; *Racine et al., 2011*), and even persist in germ-free mice (*Lino et al., 2018*), this early wave of IgM+ LLPCs is consistent with B-1-derived precursors that maintain natural antibodies (*Baumgarth, 2016*; *Baumgarth, 2011*). Indeed, LLPCs timestamped in young mice had more public clones, less diversity, and limited SHM. LLPC clones are shared across the BM and splenic compartment, suggesting these niches can be redundant and may accommodate LLPC recirculation between sites (*Benet et al., 2021*).

We find unique transcriptome and proteome expressed by endogenous LLPCs that underlie their intrinsic longevity. On the RNA level, these changes are modest (*Lam et al., 2018*), and often below detection limits for standard fold change cut-offs, which may reflect LLPC heterogeneity (*Liu et al., 2022*). However, these small changes can reflect larger changes in protein levels, as we see for CXCR4, suggesting that proteomics may be a more appropriate way to study changes in PC to LLPC maturation changes. Among the GO terms and DEGs found, most are shared by LLPCs in the BM or spleen suggesting similar requirements for survival in both sites. One notable exception was cluster 1 of DEGs among splenic LLPCs, which were associated with IgM-specific factors such as complement receptors, consistent with recent studies (*Bohannon et al., 2016*; *Higgins et al., 2022*).

A major goal here was to address how LLPCs maintain their survival. Mechanistically, is their lifetime intrinsically regulated (*Robinson et al., 2022*) or by competition for a limited niche (*Radbruch et al., 2006*)? Our imaging data supports the later model, as LLPCs are preferentially arrested and are more enriched in PCs' clusters. Previously, we reported that cluster formation is dependent on hematopoietic-derived APRIL, suggesting it may be enriched in these sites (*Benet et al., 2021*). Thus, both LLPCs and bulk PC (containing short-lived cells) may share and compete for the same cell-extrinsic cues. However, as LLPCs do not express higher levels of APRIL receptors, changes in adhesion receptors may help LLPCs preferentially dock at these niches leading to advantages in survival over newly minted PCs. Additionally, as PCs express different levels of homing and adhesion molecules according to isotype, dynamics and positioning in the BM should vary leading to distinct decay rates.

Among these LLPC-intrinsic factors, we found CXCR4 plays a direct functional role in PC survival. Our model directly targets CXCR4 during the PC stage, in an inducible fashion, in contrast with a previous study (*Nie et al., 2004*) that used a constitutive deletion of CXCR4 in B cell lineage and found no effect on humoral immunity. While it is tempting to simply conclude that increased CXCR4 expression by LLPCs directly leads to cell-intrinsic arrest in BM niches, we have also showed that CXCR4 promotes BM PC motility, and inhibitors quickly perturb PC motility (*Benet et al., 2021*). In vitro, LLPCs were unresponsive to CXCL12 chemotaxis (data not shown). Moreover, gain-of-function alleles of CXCR4 also lead to shortened humoral responses but more total PCs (*Biajoux et al., 2016*). Thus, there may be more complexity to CXCR4 function on PCs in the BM to unravel.

With age, both the number and maturation of PCs increases within the BM (*Pioli et al., 2019*), which likely affects competition. By intravital imaging, we see that overall polyclonal PC speeds increase in older mice (*Benet et al., 2021*), which is likely due to faster motility of short-lived PCs (*Koike et al., 2023*) with no sites to dock. Moreover, with aging, decreases in survival factors like APRIL (*Pangrazzi et al., 2017*) may further limit the survival of newly minted PC leading to weakened and shortened serological responses, correlating to what is seen in older adults (*Wagner and Weinberger, 2020*; *Fedele et al., 2022*).

## Materials and methods

### Mice

Prdm1-EYFP (*Fooksman et al., 2010*) (or Blimp1-YFP) BAC transgenic mice were generated previously and can also be obtained from the Jackson Laboratory (#8828). *Rosa26*$^{CAG-LSL-tdTomato}$ (Ai14, #7914) (*Madisen et al., 2010*), *Rosa26*$^{LSL-EYFP}$ (*Srinivas et al., 2001*) (#6148), and *Cxcr4*$^{fl/fl}$ [18] (#8767) C57BL/6 (CD45.2) and B6-Ly5.1/Cr (CD45.1) mice were purchased from Charles River. All mice were housed in groups of two to five animals per cage in SPF facilities at Albert Einstein College of Medicine. The animal protocol in this study was approved by Albert Einstein College of Medicine Institutional Animal Care Use Committee (IACUC), protocol #0000-1021. For PC turnover experiments, both females

and males that are young (6–8 weeks of age) or middle-aged (20–24 weeks of age) were used. For mixed BM chimera experiments, 6- to 8-week-old sex-matched mice were used as hosts, and 16- to 24-week-old WT or CXCR4$^{cKO}$ mice were used as donors.

BEC mouse was constructed using CRISPR-Cas9 technology on the C57BL/6 background by knocking-in Cre$^{ERT2}$-IRES-TdTomato cassette downstream of the exon 6 of Prdm1 locus, targeted with one single guide RNA (tctgtgggcagaaacccgcg). Founders and F1 progenies were genotyped by PCR using primers (Integrated DNA Technologies) targeting *Prdm1* genomic region (5'-ggcaagatcaagtatgagtg c-3', Forward) and IRES sequence (5'-gccaaaagacggcaatatgg-3', Reverse). This mouse line was backcrossed to C57BL/6 for at least three generations. Since BEC is a knock-in knockout allele, only heterozygotes were used for all experiments. Mice are available upon request.

## Generation of mixed BM chimera

Six- to 8-week-old CD45.1 recipient mice were lethally irradiated (950 RAD) and reconstituted with 7–8 × 10$^6$ 50:50 mixture of WT:CXCR4$^{cKO}$ total BM cells, and allowed to recover for 8 weeks with sulfamethoxazole and trimethoprim (ANI Pharmaceutials) added to the drinking water (1:50 vol/vol) in the first 2 weeks post reconstitution.

## Immunizations and treatments

For hapten-protein conjugate immunizations, WT or CXCR4$^{cKO}$ mice were immunized intraperitoneally with 50 µg of NP$_{(32)}$-KLH (Biosearch Technologies) in PBS emulsified with alum (Imject Alum; Thermo Fisher Scientific) at 2:1 vol:vol ratio in 150 µl volume. For PC turnover experiments, 4 mg tamoxifen (MilliporeSigma) were administered by oral gavage per mouse for 3 consecutive days. For intratibial injection experiments, 5 µg of 4-hydroxytamoxifen (MilliporeSigma) in 10 µl 5% ethanol (diluted with PBS) was given through shaved knee joint into the tibia using 29 G insulin syringes, and 2.5 µg PTX (MilliporeSigma) in 100 µl volume PBS were intravenously (i.v.) injected to recipient mice. For glucose uptake experiments, 50 µg 2-NBDG (Thermo Fisher Scientific) in 100 µl volume PBS was i.v. injected into mix chimeric mice for exactly 15 min before sacrifice.

## Flow cytometry

Single-cell suspensions of BM and spleen were resuspended in PBS containing 0.5% BSA and 1 mM EDTA and filtered through a 70 µm nylon mesh. Cells were first stained with LIVE/DEAD Fixable Aqua Dead Cell Stain Kit (Invitrogen). Then they were blocked with anti-CD16/32 (2.4G2, Bio X Cell) and stained for surface proteins with a combination of antibodies on ice for 30 min, and analyzed on Cytek Aurora (Cytek Biosciences). Single stains for YFP and TdTomato were prepared using blood cells from Blimp1-YFP mouse and OT-II TdTomato mouse, and all other stains were made by staining WT BM cells with individual fluorescently labeled antibody. Compensations were done by automatic live unmix in SpectroFlo software (Cytek Biosciences) during acquisition, followed by manual adjustment of the compensation matrix. To ensure the accuracy of the manual changes in the compensation matrix, day 5 middle-aged BEC-YFP BM cells were stained with each panel antibody separately to control for the matrix for each marker accordingly such that each stain consists of a basic panel including YFP, TdTomato, CD138-APC, B220-APCCy7, live/dead-CD4/8-BV510, and one of panel markers. Then, for each timepoint (except day 5) analyzed in the PC timestamping experiments, a day 5 middle-aged BEC-YFP mouse stained with all panel antibodies was included as a compensation control. The profiling of the total 19 markers were done by splitting into three subpanels with each panel sharing CD138-APC, B220-APCCy7, and live/dead-CD4/8-BV510. The antibody dilution was determined by titrating absolute amount (in µg) per million total BM cells. For intracytoplasmic (4-hydroxy-3-nitrophenyl)acetyl (NP) staining in PCs, surface-stained cells were fixed and permeabilized using BD Cytofix/Cytoperm Fixation/Permeabilization kit (BD Biosciences), followed by NP-BSA-Fluorescein (Biosearch Tech) staining in 1:200 dilution for 1 hr at 4°C.

Anti-B220 (RA3-6B2), Bcl-2 (BCL/10C4), CD4 (GK1.5), CD8 (53-6.7), CD37 (Duno85), CD44 (IM7), CD45.2 (104), CD48 (HM48-1), CD81 (Eat-2), and CD98 (RL388) were purchased from BioLegend. CD28 (37.51), CD53 (OX-79), CD79b (HM79B), CD93 (AA4.1), CD126 (D7715A7), CD138 (281-2), CD147 (RL73), CD184 (2B11), CD267 (8F10), CD268 (7H22-E16), CD319 (4G2), CD326 (G8.8) were purchased from BD Biosciences. CD3e (145-2C11), CD45.1 (A20), and CD49d (R1-2) were purchased

from Fisher Scientific. CD269 (REA550) was purchased from Miltenyi Biotec. Mcl-1 (D2W9E) was purchased from Cell Signaling Technology.

## In vitro assays

For NP-binding ELISA of mouse serum, high-binding 96-well plates (Corning Costar) were coated with 2 µg/ml NP-OVA (Biosearch Tech) in 50 µl volume bicarbonate/carbonate binding buffer (Abcam) overnight at 4°C. Parafilm were used to minimize evaporation of coating buffer inside the plate. Then coating buffer were removed and the plate was blocked with 200 µl PBS containing 1% BSA per well for 2 hr at room temperature (RT). After removing blocking buffer, serum samples were added in 50 µl volume with starting dilution at 1:4000 (vol:vol in blocking buffer) for four serial twofold dilutions in triplicates, and anti-NP standard antibody (9T13) was added in 50 µl volume with starting concentration at 1 µg/ml for eight serial twofold dilutions in duplicates, followed by incubation for 2 hr at RT. The plates were washed four times with PBS containing 0.05% Tween (PBST) before adding 50 µl peroxidase goat anti-mouse IgG-HRP (Jackson ImmunoResearch) at 1:5000 dilution (vol:vol in blocking buffer) per well for 1 hr at RT. The plates were again washed four times with PBST, followed by adding 50 µl TMB substrate (MilliporeSigma) for 5–10 min at RT, which is stopped by adding 25 µl sulfuric acid (Thermo Fisher Scientific). The plates were read by EMax Plus microplate reader (Molecular Devices) at 450 nm wavelength using SoftMax Pro 7 software.

## Multiphoton intravital imaging and analysis

Surgical preparation for BM IT imaging was done as previously described (Benet et al., 2021). Mice were anesthetized using isoflurane gas during imaging process for 4–5 hr. Z-stack images for multiple regions of tibia were collected sequentially and stitched together either before or after long-term steady-state intravital imaging using Olympus software. All imaging was performed using an Olympus FVE-1200 upright microscope, 25×1.04 NA objective, and Deepsee MaiTai Ti-Sapphire pulsed laser (Spectra-Physics) tuned to 920 nm. To maintain mouse body temperature and limit room light exposure, the microscope was fitted with custom-built incubator chamber and heated 37°C platform. Time lapses were conducted every 3 min as 100–120 µm deep Z-stacks (5 µm or 3 µm steps) with 1× zoom and with 512×512 X-Y resolution. All image analysis was conducted using Imaris software 9.3 (Bitplane) to detect and track LLPCs (YFP+TdTomato^bright) and bulk PCs (YFP+TdTomato^dim) in young and middle-aged mice and to correct drift. A ratioed channel (YFP over TdTomato, ch2/ch3) was created together with background subtraction from infrared channel (ch4) to separate LLPCs from bulk PCs. A threshold of 1.02 in track intensity mean of the ratioed channel was used so that LLPC with higher TdTomato expression would exhibit lower value below the threshold whereas bulk PCs expressing lower TdTomato level would be higher value above the threshold.

## Nearest neighbor analysis

LLPCs (YFP+TdTomato^bright) and bulk PCs (YFP+TdTomato^dim) in stitched Z-stack images were detected as described above. The 2D position coordinates (X and Y) were generated from Imaris built-in spot's function. Nearest neighbor analysis program was created in Fortran using high-performance computing. The average distance between individual LLPC spots and 20 nearest total PC spots (combining both LLPCs and bulk PCs), and between individual bulk PC spots and their 20 nearest total PC spots were calculated by the program. Then both LLPC spots and bulk PC spots were randomly picked and the sample size for each subset was determined using 95% confidence level, 5% margin of error, and total number of spots from each mouse inputted as population size. The random picking process was iterated twice per subset. The scripts were executed using a Fortran compiler (cygwin). All code of the data analysis and workflow can be viewed as text document files provided at GitHub (https://github.com/davidfooksman/nearest-neighbor; copy archived at Fooksman, 2024).

## RNA isolation and quantitative real-time RT-PCR

At least 20,000 LLPCs (YFP+TdTomato+) and 80,000 bulk PCs (YFP-TdTomato+) from BM or spleen were sorted using Aria III (BD) for total RNA extraction using RNeasy Plus Mini Kit (QIAGEN) according to the manufacturer's protocol. 30 µl RNase-free water were loaded to the spin column membrane twice to reach higher RNA concentration. 4 µl RNA samples were used for reverse transcription using High-Capacity RNA-to-cDNA Kit (Applied Biosystems) according to the manufacturer's protocol. 2 µl

cDNA from each sample were used for real-time PCR using TaqMan Universal Master Mix II with UNG (Applied Biosystems) according to the manufacturer's protocol. Predesigned TaqMan assays for Actb (Mm02619580_g1) and Cxcr4 (Mm01996749_s1) were purchased from Thermo Fisher Scientific.

## Bulk RNA-seq cDNA library preparation

BM and splenic PCs were isolated and enriched using CD138$^+$ Plasma Cell Isolation Kit (Miltenyi Biotec), and stained for CD4 (GK1.5) and CD8 (53-6.7) to dump TdTomato$^+$ T cells and DAPI for excluding dead cells before sorting on Aria III (BD) or MoFlo XDP (Beckman Coulter) for RNA extraction. ~1000 CD4$^-$CD8$^-$DAPI$^-$TdTomato$^+$ cells from each enriched samples were sorted into a PCR tube (USA Scientific) containing 0.5 μl 10× reaction buffer and half the final volume of nuclease-free water provided in SMART-Seq v4 Ultra Low Input RNA Kit for Sequencing (Takara Bio), and subsequent processes were following the manufacturer's protocol. All mixing steps were done by pipetting up and down five to six times. ERCC RNA Spike-In Control Mixes (1:5000) (Life Technologies) were added to sorted cells together with lysis buffer. Purification of amplified cDNA was done using Agencourt AMPure XP Kit (Beckman Coulter) on a magnetic separation rack for 1.5 ml tubes (New England Biolabs). The concentration of purified cDNA was determined using Qubit 1× dsDNA HS Assay Kit (Thermo Fisher Scientific) on a Qubit 2.0 fluorometer (Thermo Fisher Scientific). The quality of purified cDNA was verified on a 2100 Bioanalyzer (Agilent Technologies), and the average cDNA fragment size for a typical PC sample was peaked at approximately 600 bp following a normal distribution pattern. Finally, sequencing adaptors were added using Nextera XT DNA Library Preparation Kit (Illumina) following the manufacturer's protocol. The library containing all samples was manually mixed to ensure equal final concentration of all samples, and sent for next-generation deep sequencing by Genewiz/Azenta using NovaSeq S4 lane machine (Illumina) to reach an average of 50 million reads per sample.

## RNA-seq data processing and analysis

RNA-seq reads were aligned to the mouse genome (mm39/GRCm39) using STAR aligner (v2.6.1b) (*Dobin et al., 2013*). Counts for individual genes were quantified using the RSEM program (v1.3.1) (*Li and Dewey, 2011*). Differential expression was computed using the DESeq2 (v 1.26.0), from pair-wise comparisons at adjusted p-value <0.05 (without additional fold change threshold) were collected, clustered by k-mean clustering (k=5), and used for GO enrichment analysis with the https://patherdb. org/ server. Enriched GO at adjusted p<0.05 were obtained and then DEGs in biologically relevant GO terms for each of the six comparisons were subject to over-representation analysis by the Fisher's test, with the results shown as bubble plots. Raw data and processed files were uploaded to the NCBI server (GSE221251).

## Transmission electron microscopy

PCs are isolated and enriched the same way as mentioned in the 'Bulk RNA-seq cDNA library preparation' section. 4000–8000 LLPCs (YFP$^+$TdTomato$^+$) and 40,000–200,000 bulk PCs (YFP$^-$TdTomato$^+$) were subsequently collected by sorting into a 500 μl low adhesion microcentrifuge tubes (USA Scientific) using Aria III (BD) or MoFlo XDP (Beckman Coulter), and 5–6 million sheep red blood cells (Innovative Research) were added to the same tube to provide contrast to the PCs, as previously described (*Scotton et al., 2002*), which were pelleted by centrifugation at 350×*g* for 5 min at RT. The supernatant was removed by aspiration, and the fixative containing 2.5% glutaraldehyde in 0.1 M cacodylate (prewarmed at RT) was gently added by layering on top of the residual volume of buffer including the cell pellets for 15 min at RT. Samples were postfixed with 1% osmium tetroxide followed by 2% uranyl acetate, dehydrated through a graded series of ethanol, and embedded in LX112 resin (LADD Research Industries, Burlington, VT). Ultrathin sections were cut on a Leica Ultracut UC7, stained with uranyl acetate followed by lead citrate and viewed on a JEOL 1400 Plus transmission electron microscope at 120 kV.

## BCR repertoire analysis

BCR clones were inferred from RNA-seq data, individually for each sample, using MIXCR v4.0.0b (*Bolotin et al., 2015*) using the following commands:

```
mixcr align -s mmu -p kAligner2
mixcr assemble --write-alignments
```

```
mixcr assembleContigs
mixcr exportClones -c IG -p fullImputed
```

The resulting clone files were pre-processed using a custom Python script to separate IGH, IGK, and IGL clones and to remove small clones of size <10. The resulting datasets were processed using the R package immunarch (https://immunarch.com/). Each repertoire (IGH, IGK, IGL) was loaded using repLoad. Diversity (Chao1) statistics were calculated using repDiversity and repertoire overlaps using repOverlap. For SHM estimates, we used custom R scripts. We filtered out fragmented (lists of sequences with commas), then processed the resulting sequences through IMGT-High V-Quest, which identifies mutations with respect to the closest germline sequence (from IMGT file '8_V-REGION-nt-mutation-statistics.txt'). To avoid double-counting of mutations within a clone, we selected a random sequence from each clone (most clones only had one sequence), then calculated the mean mutation frequency per sequence (number of mutations/V gene length), and then aggregated these to calculate a mean for each IGHV gene allele within each sample (e.g. the mean for all clones assigned to the IGHV8-9*01 allele). Pair-wise statistical comparisons between the samples were performed using a paired t test based on matching alleles (alleles that did not match were not used). Benjamini-Hochberg corrected p-values were calculated using the R function p.adjust with the argument method='BH'.

## Quantification and statistical analyses

Statistical tests were performed using GraphPad Prism (v7 and v8). Specific tests used in each figure are provided in the figure legends with asterisks for statistical significance (*, p-value≤0.05; **, p-value≤0.01; ***, p-value≤0.001; ****, p-value≤0.0001) or 'ns' denoting comparisons that are not statistically significant. Data are presented as the mean ± SD or mean ± SEM. For PC half-life ($t_{1/2}$) calculation, the procedure was done exactly as previously described (*Xu et al., 2020*). For RT-qPCR analysis, $2^{-\Delta\Delta Ct}$ method was used to calculate the fold change in *Cxcr4* gene expression relative to the expression of housekeeping gene *Actb* in WT samples.

## Acknowledgements

We would like to thank Dr. Yongwei Zhang, Einstein Transgenic Facility, for helping construct BEC mice, Dr. Xusheng Zhang for help with generating surface marker heatmap, Einstein Flow Cytometry Core for FACS sorting, Einstein Genomics Core for bioanalysis, Einstein Analytical Imaging Facility for help with EM imaging and analysis. This work was supported by R01HL141491 (ZJ, LO, RP, DF), Irma T Hirschl/Monique Weill-Caulier Trusts Research Award (DF), R01AI132633 (MD, KC) with support from the Albert Einstein NCI Cancer Center grant P30CA013330, SIG #1S10OD016214-01A1. We thank Leslie Cummins for help analyzing and preparing EM images. We thank Dr. Gregoire Lauvau for comments on the manuscript. We would like to dedicate this study to the memory of our friend and co-author, Dr. Thomas MacCarthy.

## Additional information

### Funding

| Funder | Grant reference number | Author |
|---|---|---|
| National Institutes of Health | R01HL141491 | Zhixin Jing<br>Luis Ovando<br>Rosa Park<br>David Fooksman |
| National Institutes of Health | R01AI132633 | Megan Demouth<br>Kartik Chandran |
| Irma T. Hirschl Trust | Irma T Hirschl/Monique Weill-Caulier Trusts Research Award | David Fooksman |
| Monique Weill-Caulier Trust | Irma T Hirschl/Monique Weill-Caulier Trusts Research Award | David Fooksman |

| Funder | Grant reference number | Author |
|---|---|---|
| Albert Einstein Cancer Center | P30CA013330 | David Fooksman |

The funders had no role in study design, data collection and interpretation, or the decision to submit the work for publication.

## Author contributions

Zhixin Jing, Investigation, Writing - original draft, Writing - review and editing; Phillip Galbo, Data curation, Software; Luis Ovando, Rosa Park, Investigation, Methodology; Megan Demouth, Methodology; Skylar Welte, Formal analysis; Kartik Chandran, Supervision; Yinghao Wu, Thomas MacCarthy, Deyou Zheng, Data curation, Software, Formal analysis; David Fooksman, Conceptualization, Resources, Formal analysis, Supervision, Funding acquisition, Validation, Investigation, Visualization, Methodology, Writing - original draft, Project administration, Writing - review and editing

## Author ORCIDs

Zhixin Jing http://orcid.org/0000-0003-1118-7432
Phillip Galbo http://orcid.org/0000-0002-5391-598X
Luis Ovando http://orcid.org/0000-0002-6449-7365
Rosa Park http://orcid.org/0000-0003-4356-8838
Deyou Zheng http://orcid.org/0000-0003-4354-5337
David Fooksman http://orcid.org/0000-0001-7601-4255

## Ethics

All mice were housed in groups of 2-5 animals per cage in SPF facilities at Albert Einstein College of Medicine. The animal protocol in this study was approved by Albert Einstein College of Medicine Institutional Animal Care Use Committee (IACUC), protocol #0000-1021.

Reviewer #1 (Public Review): https://doi.org/10.7554/eLife.89712.3.sa1
Reviewer #2 (Public Review): https://doi.org/10.7554/eLife.89712.3.sa2
Reviewer #3 (Public Review): https://doi.org/10.7554/eLife.89712.3.sa3
Author response https://doi.org/10.7554/eLife.89712.3.sa4

# Additional files

## Supplementary files

• MDAR checklist

## Data availability

All code of the data analysis and work flow can be viewed as text document files provided at GitHub (copy archived at *Fooksman, 2024*). Sequencing data and processed files were uploaded to the NCBI server (GSE221251).

The following dataset was generated:

| Author(s) | Year | Dataset title | Dataset URL | Database and Identifier |
|---|---|---|---|---|
| Fooksman D, Jing Z, Zheng D, Galbo P, MacCarthy T | 2024 | Maturation of Long Lived Plasma cells | https://www.ncbi.nlm.nih.gov/geo/query/acc.cgi?acc=GSE221251 | NCBI Gene Expression Omnibus, GSE221251 |

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
