## [Editor Report · eLife assessment]

Despite the importance of long-lived plasma cells (LLPCs), particularly for the infection and vaccination field, it is still unclear how they acquire their longevity. With a **solid** genetic approach, the authors demonstrate quite **convincingly** a requirement for chemokine/chemokine receptor-mediated interaction in LLPC longevity. The data are very **valuable** for the development of new types of vaccines.

---

## [Referee Report · Reviewer #1 (Public Review)]

The mechanisms underlying the generation and maintenance of LLPCs have been one of the unresolved issues. In the last few years, several groups have independently generated new genetic tools or models and addressed how LLPCs are generated or maintained in homeostatic conditions or upon immunization or infection. Here, Jing et al. have also established a new PC time stamping system and tried to address the issues above. The authors have found that LLPCs accumulated in the BM PC pool, along with aging, and that LLPCs had unique sufacetome, transcriptome, and BCR clonality. These observations have already been made by other groups (Xu et al. 2020, Robinson et al. 2022, Liu et al. 2022, Koike et al. 2023, Robinson et al. 2023, plus Tellier et al., 2024), therefore it is hard to find significant conceptual advances there. In my opinion, however, genetic analysis of the role of CXCR4 on PC localization or survival in BM (Figure 4 and 5) provided new aspects which have not been addressed in previous studies. Importantly, CXCR4 was required for the maintenance of plasma cells in bone marrow survival niches, conditional loss of which led to rapid mobilization from the bone marrow, reduced plasma cell survival, and reduced antibody titer. Thus, these data suggest that CXCR4-CXCL12 axis is not only important for plasma cell recruitment to the bone marrow but also essential for their lodging on the niches. I think the study is of high quality and the findings should be widely shared in the field.

---

## [Referee Report · Reviewer #2 (Public Review)]

In this study by Jing, Fooksman, and colleagues, a Blimp1-CreERT2-based genetic tracing study is employed to label plasma cells. Over the course of several months post-tamoxifen treatment, the only remaining labeled cells are long-lived plasma cells. This system provides a way to sort live long-lived plasma cells and compare them to unlabeled plasma cells, which contain a range of short-to-long-lived cells. From this analysis, several observations are made: (1) the turnover rate of plasma cells is greater in the spleen than in the bone marrow; (2) the turnover rate is highest early in life; (3) subtle transcriptional and cell surface marker differences distinguish long- from shorter-lived plasma cells; (4) long-lived plasma cells in the bone marrow are sessile and localize in clusters with each other; (5) CXCR4 is required for plasma cell retention in these clusters and in the bone marrow; (6) Repertoire analysis hints that the selection of long-lived plasma cells is not random for any cell that lands in the bone marrow.

Strengths:

(1) The genetic timestamping approach is a clever and functional way to separate plasma cells of differing longevities.

(2) This approach led to the identification of several markers that could help prospective separation of long-lived plasma cells from others.

(3) Functional labeling of long-lived plasma cells allowed for a higher resolution analysis of transcriptomes and motility than was previously possible.

(4) The genetic system allowed for a revisitation of the importance of CXCR4 in plasma cell retention and survival.

Weaknesses:

(1) Most of the labeling studies, likely for practical reasons, were done on polyclonal rather than antigen-specific plasma cells. The triggers of these responses could vary based on age at the time of exposure, anatomical sites, etc. How these differences might influence markers and transcriptomes, independently of longevity, is not completely known.

(2) The fraction of long-lived plasma cells in the unlabeled fraction varies with age, potentially diluting differences between long- and short-lived plasma cells.

(3) The authors suggest their data favors a model by which plasma cells compete for niche space. Yet there is no evidence presented here that these niches are limiting. While a finite number of plasma cells may occupy a single niche (Figure 2), it may be that these niches overall are abundant in the bone marrow and do not restrict LLPC numbers. Robinson...Tarlinton and colleagues (Immunity, 2023) in fact provide experimental evidence against an extrinsic limit.

(4) The functional importance of the observed transcriptome differences between long- and shorter-lived plasma cells is unknown. An assessment as to whether these differences are conserved in human long- and short-lived bone marrow plasma cells might provide circumstantial supporting evidence that these changes are important for longevity.

---

## [Referee Report · Reviewer #3 (Public Review)]

Summary:

Long-lived PCs are maintained in a CXCR4-dependent manner.

Strengths:

The reporter mice for fate-mapping can clearly distinguish long-lived PCs from total PCs and greatly contribute to the identification of long-lived PCs.

---

## [Author Response]

The following is the authors’ response to the original reviews.

**eLife assessment**
Despite the importance of long-lived plasma cells (LLPCs), particularly in the vaccination field, their natures are still unclear. In this valuable manuscript, as a first step towards clarifying these natures, the authors used a solid genetic approach (time-stamping one) and successfully labelled only functional LLPCs. Although four groups have already published data by the same genetic approach, the authors' manuscript includes additional significant findings in the LLPC field.
**Public Reviews:**

**Reviewer #1 (Public Review):**
The mechanisms underlying the generation and maintenance of LLPCs have been one of the unresolved issues. Recently, four groups have independently generated new genetic tools that allow fate tracing of murine plasma cells and have addressed how LLPCs are generated or maintained in homeostatic conditions or upon antigen immunization or viral infection. Here, Jing et al. have established another, but essentially the same, PC time stamping system, and tried to address the issues above. The question is whether the findings reported here provide significant conceptual advances from what has already been published.(1) Some of the observations in this manuscript have already been made by other studies (Xu et al. 2020, Robinson et al. 2022, Liu et al. 2022, Koike et al. 2023, Robinson et al. 2023). In my opinion, however, genetic analysis of the role of CXCR4 on PC localization or survival in BM (Figure 5) was well performed and provided some new aspects which have not been addressed in previous reports. The motility of CXCR4 cKO plasma cells in BM is not shown, but it could further support the idea that reduced mobility or increased clustering is required for longevity.(2) The combination of the several surface markers shown in Figure 3&4 doesn't seem to be practically applicable to identify or gate on LLPCs, because differential expression of CD81, CXCR4, CD326, CD44, or CD48 on LLPCs vs bulk PCs was very modest. EpCAMhi/CXCR3-, Ly6Ahi/Tigit- (Liu et al. 2022), B220lo/MHC-IIlo (Koike et al. 2023), or SLAMF6lo/MHC-IIlo (Robinson et al. 2023) has been reported as markers for LLPC population. It is unclear that the combination of surface markers presented here is superior to published markers. In addition, it is unclear why the authors did not use their own gene expression data (Fig.6), instead of using public datasets, for picking up candidate markers.

In terms of the utility of these markers, we agree they are not sufficient to distinguish bona fide LLPCs but they did enrich for LLPCs by 6-fold (Figure 3). In the other studies cited, LLPCs are enriched in those gates but not exclusively found in the gates, suggesting some plasticity. In terms of how they were chosen, we conducted the flow surface studies in parallel and prior to completing the gene expression studies, thus, they were not available in time to be useful for the longitudinal studies. As this was not the major findings of the paper, we have reduced emphasis on this section, and moved some of the data to Figure S2.

**Reviewer #2 (Public Review):**
In this study by Jing, Fooksman, and colleagues, a Blimp1-CreERT2-based genetic tracing study is employed to label plasma cells. Over the course of several months post-tamoxifen treatment, the only remaining labeled cells are long-lived plasma cells. This system provides a way to sort live long-lived plasma cells and compare them to unlabeled plasma cells, which contain a range of short-to-long-lived cells. From this analysis, several observations are made: (1) the turnover rate of plasma cells is greater in the spleen than in the bone marrow; (2) the turnover rate is highest early in life; (3) subtle transcriptional and cell surface marker differences distinguish long- from shorter-lived plasma cells; (4) long-lived plasma cells in the bone marrow are sessile and localize in clusters with each other; (5) CXCR4 is required for plasma cell retention in these clusters and in the bone marrow; (6) Repertoire analysis hints that the selection of long-lived plasma cells is not random for any cell that lands in the bone marrow.Strengths:(1) The genetic timestamping approach is a clever and functional way to separate plasma cells of differing longevities.(2) This approach led to the identification of several markers that could help prospective separation of long-lived plasma cells from others.(3) Functional labeling of long-lived plasma cells allowed for a higher resolution analysis of transcriptomes and motility than was previously possible.(4) The genetic system allowed for a revisitation of the importance of CXCR4 in plasma cell retention and survival.Weaknesses:(1) Most of the labeling studies, likely for practical reasons, were done on polyclonal rather than antigen-specific plasma cells. The triggers of these responses could vary based on age at the time of exposure, anatomical sites, etc. How these differences might influence markers and transcriptomes, independently of longevity, is not completely known.(2) The fraction of long-lived plasma cells in the unlabeled fraction varies with age, potentially diluting differences between long- and short-lived plasma cells.(3) The authors suggest their data favors a model by which plasma cells compete for niche space. Yet there is no evidence presented here that these niches are limiting.

In Figure 2, we provide important evidence that LLPCs are enriched in PC clusters, and are less motile, suggesting they occupy a unique niche compared to bulk PCs in the bone marrow. But we agree it does not clarify if that niche is limited.

(4) The functional importance of the observed transcriptome differences between long- and shorter-lived plasma cells is unknown. An assessment as to whether these differences are conserved in human long- and short-lived bone marrow plasma cells might provide circumstantial supporting evidence that these changes are important for longevity.
**Reviewer #3 (Public Review):**
The valuable work shows some unique characteristics of long-lived PCs in comparison with bulk PCs. In particular, the authors clearly indicated the dependency of CXCR4 in PC longevity and provided a deal of resource of PC transcriptomes. Though CD93 is known as a marker for long-lived PCs, the authors can provide more data related to CD93.Summary:Long-lived PCs are maintained with low motility and in a CXCR4-dependent manner.Strengths:The reporter mice for fate-mapping can clearly distinguish long-lived PCs from total PCs and greatly contribute to the identification of long-lived PCs.Weaknesses:The authors are unable to find a unique marker for long-lived PCs
**Recommendations for the authors:**

**Reviewer #1 (Recommendations For The Authors):**
(1) Given the author's expertise, I suggest investigating the motility of CXCR4 cKO plasma cells in BM.

Thank you for the suggestion. This work would certainly fit in with the theme of the paper. We tried to measure this using the BEC Rosa-LSL-YFP Cxcr4f/f system after tamoxifen treatment but unfortunately, these PCs leave the BM concurrently as they turn on YFP expression from the Rosa26 locus, making it impossible to capture the change in motility. This is also evident in our data in updated Figure 5 which shows that intratibial injection of 4HO-Tamoxifen causes rapid mobilization of CXCR4KO PCs from the tibia within 1 day. We tried to breed other models that would allow us to visualize these early events, which were unsuccessful, and also responsible for the long delay in resubmission.

(2) Expression of CD81, CXCR4, CD326, CD44, or CD48 was not different enough to distinguish LLPCs from bulk PCs (Figure 3B). The caveat is that bulk PCs also contained a significant frequency of LLPCs, which would make the difference in expression levels smaller. I suggest looking at the expression of these molecules on newly generated PCs, soon after protein immunization, for example.

This would be a separate issue, when they begin to express the LLPC phenotype, and definitely worthwhile in future studies.

**Reviewer #2 (Recommendations For The Authors):**
(1) Related to the above public comment #4, I would recommend looking at Halliley et al., Immunity, 2015 to see if some of the same LLPC transcriptional and marker differences can be observed between CD19+ and CD19- plasma cells in the human marrow.

Thank you for the suggestion to do a human correlation. It is unclear what conclusions we can draw from overlapping or non-overlapping patterns, on their own.

(2) For CD93, since it is bimodal, it may be better to express this as % positive rather than fold changes in MFI as in Figure 3.

We have updated Figure 3C to include %positive as suggested. Fold changes were moved to Figure S2.

**Reviewer #3 (Recommendations For The Authors):**
The valuable work shows some unique characteristics of long-lived PCs in comparison with bulk PCs. In particular, the authors clearly indicated the dependency of CXCR4 in PC longevity and provided a deal of resources of PC transcriptomes. Though CD93 is known as a marker for long-lived PCs, the authors can provide more data related to CD93.Major points:The authors show data that some bulk PCs express CD93 lower. Are CD93low bulk PCs are higher motile in the BM compared to CD93high? Are CD93low highly mutated in the Ig gene? Do CD93high bulk PCs have similar transcriptome to long-lived PCs on some representative genes?

Although we do not have data here, the difference between CD93high cells and CD93low cells are likely to be small since labeled PCs were observed to express higher CD93 surface level as early as day 5 in BM and SP shown in updated Figure 3C. Thus, while CD93 is strongly enriched in LLPCs, it cannot be used as a single marker to sufficiently isolate LLPCs, which would make it very difficult to detect changes in motility, mutation of Ig gene, and gene expression.

Minor points:(1) In the title, the authors describe that surface receptor expression support PC-intrinsic longevity. The surface receptor is only CXCR4. The ambiguous description confuses the readers.

While CXCR4 was shown functionally to be involved, we found multiple surface receptors are differentially expressed in LLPCs.

(2) The abbreviations of 'bone marrow' and 'BM' should be unified.(3) In Fig. 7C, the bars for comparison are unclear. What dots are compared?

Bars are comparing day 90 middle aged to day 5 controls, as there were only n=2 for some day 90 young mice samples for all internally pared comparisons.

(4) The explanation about Fig.7I can't be understood. How are conclusions occurred from the panel?

Fig. 7I shows that of the most common public clones found (found in the most samples or mice), across all LLPC and Bulk 42 total samples, most of the hits came from LLPC samples (all colored) whereas few were from bulk PC samples (white bars), suggesting the shared repertoire is uniquely LLPC-like. These were observations drawn, but no statistical analysis was conducted here.